# RUAGO: Effective and Practical Retain-Free Unlearning via Adversarial Attack and OOD Generator

**Sangyong Lee[1], Sangjun Chung[2], Simon S. Woo[1, 2, 3] ***

[1] Department of Computer Science and Engineering, Sungkyunkwan University, South Korea
[2] Department of Artificial Intelligence, Sungkyunkwan University, South Korea
[3] Secure Machines Lab Inc, South Korea
{sang8961,hyjk826,swoo}@g.skku.edu

## Abstract

With increasing regulations on private data usage in AI systems, machine unlearning has emerged as a critical solution for selectively removing sensitive information from trained models while preserving their overall utility. While many existing unlearning methods rely on the *retain data* to mitigate the performance decline caused by forgetting, such data may not always be available (*retain-free*) in real-world scenarios. To address this challenge posed by retain-free unlearning, we introduce **RUAGO**, utilizing adversarial soft labels to mitigate over-unlearning and a generative model pretrained on out-of-distribution (OOD) data to effectively distill the original model's knowledge. We introduce a progressive sampling strategy to incrementally increase synthetic data complexity, coupled with an inversion-based alignment step that ensures the synthetic data closely matches the original training distribution. Our extensive experiments on multiple benchmark datasets and architectures demonstrate that our approach consistently outperforms existing retain-free methods and achieves comparable or superior performance relative to retain-based approaches, demonstrating its effectiveness and practicality in real-world, data-constrained environments.

## 1 Introduction

In recent years, the success of machine learning across diverse domains has been driven by the availability of massive datasets. A prominent example is GPT-4, a milestone in advancing machine learning, which was trained on approximately 5 trillion data points [1]. However, these advancements have raised serious privacy concerns due to including sensitive or unauthorized information in training data. Widely deployed models such as ChatGPT have also demonstrated risks of information leakage [2]. In response, regulations [3, 4] such as the EU/US Copyright Law [5] and the General Data Protection Regulation (GDPR) emphasize the *Right to be Forgotten*, ensuring "the data subject shall have the right to obtain from the controller the erasure of personal data concerning him or her without undue delay" [6]. Consequently, model owners, including those of AI systems, must carefully manage personal data and comply with removal requests. However, removing data may alter the original model's behavior and can lead to performance degradation. The most straightforward solution is to retrain the model without the requested data, referred to as the *forget data*, but this strategy is computationally and financially prohibitive.

To address rising privacy concerns and regulatory demands, *Machine Unlearning* (MU) [7, 8] has emerged as a framework for removing specific training data. While early work focused on simple

---

*Simon S. Woo is the corresponding author.

models [9–11], recent efforts have extended MU to deep neural networks [12–16]. In particular, MU can be interpreted as a multi-objective task, where the two main opposing goals are removing the forget data while preserving the utility of the remaining data, which we refer to as the *retain data*. Since forgetting significantly affects model behavior and performance, achieving both objectives is challenging. A further challenge is to minimize overall training costs. Although recent SOTA methods [13, 17, 18] show strong unlearning performance, they often assume full access to the original or retain data. However, such assumptions are often impractical due to storage limitations, expired permissions, or privacy constraints. To mitigate this issue, recent approaches [12, 14–16, 19, 20] explore the *retain-free* setting, using only the forget data. Although these retain-free methods have demonstrated strong performance in class-wise unlearning, a scenario involving the removal of entire information from a model, their effectiveness in instance-wise unlearning [12] remains uncertain and unexplored. Instance-wise unlearning, which targets specific samples instead of entire classes, is particularly challenging due to distributional overlap between forget and retain data.

In this work, we present a novel unlearning method for the retain-free scenario, **R**etain-free **U**nlearning via **A**dversarial attack and **G**enerative model using **O**OD training, **RUAGO** as briefly illustrated in Fig. 1. Our method investigates the critical issue of existing instance-wise unlearning methods, particularly *over-unlearning*, which occurs when excessive removal severely degrades model performance. To mitigate this, we generate adversarial probabilities and use them as soft labels for forget data. These soft labels are positioned near decision boundaries and guide the forgetting process more smoothly than deterministic hard labels, which often induce drastic parameter shifts and information loss in instance-wise scenarios.

However, relying solely on the above strategy may still degrade model generalization performance, especially in the retain-free setting, where access to retain data is restricted. To overcome this challenge, we use a generative model trained on out-of-domain (OOD) data, which can avoid training on the retain data and eliminates the privacy risks associated with using a generator trained on the original dataset. In contrast, we believe synthetic

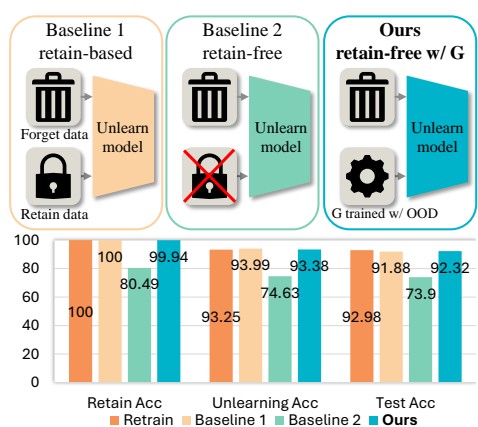

Figure 1: Conceptual overview of RUAGO. Our method, RUAGO ("retain-free w/ G"), uses a generator ('G') trained on out-of-distribution (OOD) data, requiring no retain set. It is compared with a retain-based method, Bad Teacher [17] ("Baseline 1"), and a retain-free one, Boundary Shrink [19] ("Baseline 2"). The figure shows RUAGO's performance is comparable to the retain-based baseline and superior to the retain-free one.

data generated from OOD distributions can inherently and completely mitigate privacy concerns, while effectively transferring essential knowledge from the original model, thus preserving model utility [21]. To enhance the quality of synthetic supervision, we introduce two additional strategies. First, through a VC theory-based analysis, we demonstrate that progressively increasing the model's complexity during knowledge distillation significantly improves the final model's performance and generalization stability. Therefore, we design a dynamic sample difficulty scheduler, assigning higher initial weight to simpler synthetic samples to support stable early convergence and gradually introducing more complex samples to enhance model robustness. Second, we extract key characteristics from the original model via inversion-based fine-tuning, and progressively align the generator's output distribution with the original data distribution, even without access to retain data. Consequently, after refinement, the OOD-trained generator produces samples similar to the original distribution, enabling effective knowledge transfer and preserving robust performance in retain-free conditions. We validate RUAGO through extensive experiments on four benchmark datasets and three network architectures, showing superior performance over retain-free baselines and competitive results with SOTA retain-based methods.

Our contributions are summarized as follows:

- We propose RUAGO, a novel retain-free unlearning framework that leverages adversarial soft labels to prevent over-unlearning and employs synthetic data from an OOD-trained generator to preserve model performance and protect data privacy in retain-free scenarios.

- We introduce a dynamic scheduling strategy guided by sample difficulty and grounded in VC theory, gradually increasing model complexity during distillation. We also refine the OOD-trained generator via inversion-based alignment to better match the original data distribution and improve knowledge transfer.

- Through comprehensive experiments, we show that RUAGO consistently outperforms prior retain-free methods and matches retain-based approaches across diverse datasets and architectures, highlighting its effectiveness and practical utility in realistic scenarios.

## 2 Related Work

### 2.1 Machine Unlearning

The primary objective of machine unlearning (MU) is to effectively remove information from the forget data, while maintaining the utility of the retain data [19]. We categorize MU into retain-based and retain-free methods based on whether the retain dataset is required during unlearning.

**Retain-based methods** utilize the retain data to support the unlearning process. Various methods have been proposed to enhance the efficiency of the retraining process for the retain data [7, 22, 23]. However, due to the substantial computational resources and memory requirements, many studies have focused on updating parameters within pre-trained models. Goel et al. [24] proposed two methods, retraining the last $k$ layers from scratch (EU-k) and fine-tuning the last $k$ layers (CU-k) using the retain data. Bad-T [17] and SCRUB [13] employ a teacher-student framework wherein the teacher induces forgetting through positive and negative knowledge transfer. SalUn [25] identifies and modifies weights with significant influence on the forget data, thereby effectively removing them. Despite demonstrating strong unlearning performance, these approaches require large amounts of the retain data, limiting their applicability in scenarios where access to the retain data is constrained.

**Retain-free methods** focus on unlearning without the retain data. In such scenarios, effectively forgetting specific data while maintaining overall model performance is difficult and often necessitates additional metadata. Yoon et al. [26] used model inversion to train a conditional GAN (CGAN) along with the forget data to perform unlearning, but it suffers from the complexity of model inversion and underperforms compared to recent methods. Boundary unlearning [19] employs adversarial attacks to guide the forget data toward incorrect labels near decision boundaries. SSD [14] computes the relative importance of parameters between the entire training dataset and the forget data, selectively dampening parameters. While it achieves effective unlearning without training, it requires importance information from the entire training dataset. SCAR [16] uses Mahalanobis distance to delete the influence of the forget data and maintain performance without accessing the retain data by leveraging OOD data. However, its reliance on metadata such as means and covariance matrices limits its applicability in scenarios where this information is unavailable. Moreover, these methods generally exhibit lower unlearning performance than the retain-based methods, and face additional challenges when applied to instance-wise unlearning.

### 2.2 Curriculum Learning

Curriculum Learning (CL) is a training strategy that mimics human learning by gradually adjusting the difficulty of the training process, progressing from easier to harder samples [27]. This approach facilitates stable learning by allowing models to incrementally build their understanding across various tasks [28–30]. We reinterpret the teacher-student relationship in knowledge distillation through the lens of VC theory, showing that achieving generalization in early stages with limited model capacity leads to more stable subsequent learning. Building on this theoretical insight and inspired by CL, we implement a progressive knowledge transfer strategy by assigning weights to generated images based on their difficulty, thus guiding the model from easier to harder samples. To the best of our knowledge, we are the first to utilize the concept of sample difficulty in the field of MU.

### 2.3 Data-Free Knowledge Distillation

Data-Free Knowledge Distillation (DFKD) aims to train student model without the use of original training data. The core idea of DFKD is to generate synthetic data that mimics the distribution of the original training data through a model inversion process, considering the teacher model as a

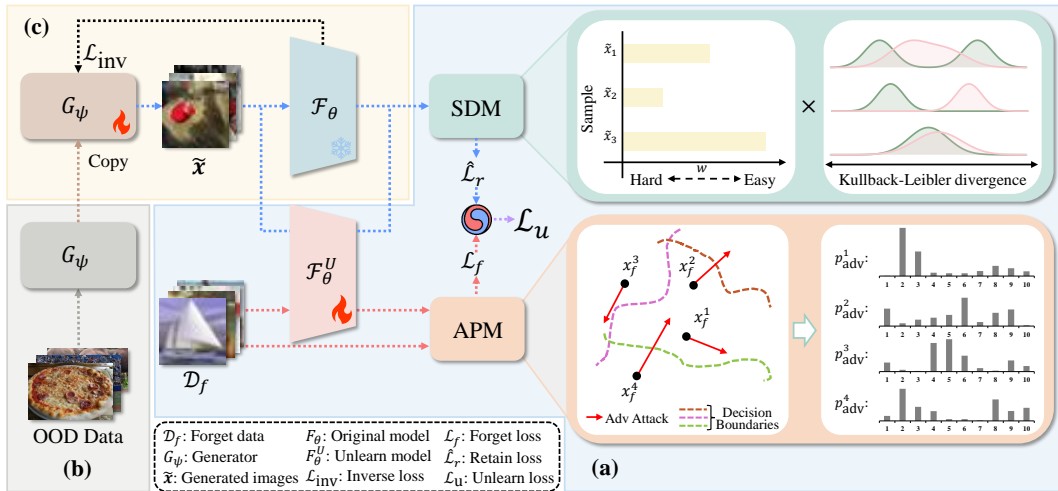

Figure 2: Overall procedure of our proposed method. (a) Unlearning the model, where the blue arrow denotes knowledge distillation from the original model via generated images with the sample difficulty module (SDM); and the red arrow denotes deletion of the forget set $\mathcal{D}_f$ via the adversarial probability module (APM). (b) Training the generator $G_\psi$ on an OOD dataset. (c) Aligning $G_\psi$ to the training data $\mathcal{D}$ via model inversion.

discriminator. This process often involves using generative models such as Generative Adversarial Networks (GANs) to generate synthetic data [31–34]. The student model is trained on these synthetic data to match the predictions of the teacher model using distillation process. In this work, we apply the model inversion strategy to the unlearning process, focusing on effectively preventing performance decline from a model without requiring access to the original training data. By leveraging synthetic data generated through a generator, our approach enables precise unlearning, addressing key challenges in real-world scenarios.

## 3 Preliminaries and Problem Statement

We first introduce notation for a supervised image-classification setting. Let $\mathcal{X} \subset \mathbb{R}^d$ be the image space and $\mathcal{Y} = \{1, \ldots, c\}$ the label set. We denote the full training dataset $\mathcal{D} = \{(x^i, y^i)\}_{i=1}^N \subseteq \mathcal{X} \times \mathcal{Y}$. Let $\mathcal{D}_t \subseteq \mathcal{X} \times \mathcal{Y}$ be an independent test set from the same distribution. We assume that $\mathcal{F}_\theta : \mathcal{X} \to \mathcal{Y}$ is a deep learning model with parameter $\theta$ trained on $\mathcal{D}$.

Machine unlearning seeks an updated model $\mathcal{F}_\theta^U$ that "forgets" the *forget set* $\mathcal{D}_f \subset \mathcal{D}$, while preserving performance on the *retain set* $\mathcal{D}_r = \mathcal{D} \setminus \mathcal{D}_f$ and on $\mathcal{D}_t$. Formally, $\mathcal{F}_\theta^U$ should behave such as $\mathcal{F}_{\theta^*}$, a model retrained from scratch on $\mathcal{D}_r$. In the instance-wise unlearning scenario, where individual examples are removed regardless of their class, rather than removing entire classes as in class-wise unlearning, the resulting class distribution of the forget set $\mathcal{D}_f$ and the retain set $\mathcal{D}_r$ remains nearly identical. This heavy overlap between $\mathcal{D}_f$ and $\mathcal{D}_r$ makes it difficult to remove information of $\mathcal{D}_f$ without inadvertently degrading performance on $\mathcal{D}_r$. We therefore evaluate $\mathcal{F}_\theta^U$ against $\mathcal{F}_{\theta^*}$ using standard accuracy metrics on $\mathcal{D}_r$ and $\mathcal{D}_t$, as well as membership inference attack (MIA) metric.

## 4 Our Approach

This section introduces RUAGO, our framework for instance-wise unlearning without access to a retain set $\mathcal{D}_r$. Instead of relying solely on the forget set $\mathcal{D}_f$, which can degrade overall performance and utility, RUAGO focuses on adversarial probabilities to guide the forgetting process and a generator pretrained on out-of-domain (OOD) data to preserve decision-boundary fidelity to the original model. To stabilize distillation from synthetic samples generated by the OOD-pretrained generator, we develop a VC-theoretic analysis of knowledge distillation and implement a dynamic sample-difficulty schedule. Additionally, we refine the generator via a model-inversion step to align its outputs with the original training distribution $\mathcal{D}$. The complete pipeline is illustrated in Fig. 2.

## 4.1 Adversarial Probability

Adversarial attacks [35, 36] inject imperceptible perturbations into images to maximize model prediction errors, thereby inducing misclassification toward the nearest alternative class in the loss landscape. While traditionally employed to evaluate model robustness or construct defenses against adversaries, these perturbations have recently been applied in machine unlearning [12, 15, 19], where adversarial labels generated by perturbations serve as alternative targets for forget-set samples $\mathcal{D}_f$. However, unlike class-wise unlearning methods, instance-wise unlearning does not aim to completely misclassify samples in $\mathcal{D}_f$ away from their original classes. Since $\mathcal{D}_f$ constitutes unseen data from the retrained model's perspective, these samples inherently carry higher ambiguity than those in the retain set, $\mathcal{D}_r$. Consequently, although the retrained model may exhibit uncertainty toward $\mathcal{D}_f$, it can still correctly predict a subset of it. Therefore, strictly assigning completely different class labels to samples in $\mathcal{D}_f$ risks causing *over-unlearning* [25]. Specifically, recall that using hard one-hot targets $q$ with cross-entropy loss, defined as $L = -\sum q \log(p)$, results in gradients $\frac{\partial L}{\partial z} = p - q$. For the targeted class $j$, this yields a gradient of $p^j - 1$, which can become large in magnitude, triggering drastic parameter updates and potentially destroying useful general features.

To circumvent this issue, we instead propose using the adversarial probability vector $\mathbf{p}_{\mathrm{adv}}$ as a *soft* target. These soft labels distribute gradient contributions more evenly, as $\frac{\partial L}{\partial z} = p - p_{\mathrm{adv}}$, thereby producing milder parameter updates and preserving the model's inherent uncertainty toward samples in the forget set. This approach effectively treats $\mathcal{D}_f$ analogously to unseen or ambiguous data, mitigating its influence without excessive parameter drift. Moreover, this design aligns with knowledge distillation principles [37], wherein soft targets convey richer probabilistic information and exhibit lower gradient variance than hard labels. Consequently, our instance-wise unlearning method not only prevents over-unlearning but also better maintains the model's overall utility.

Formally, adversarial examples and probabilities are computed as:

$$\mathbf{x}_{\mathrm{adv}} = \mathbf{x} + \underset{|\delta| \leq \epsilon}{\arg\max} \, \mathcal{L}\big(\mathcal{F}(\mathbf{x} + \delta), \mathbf{y}\big), \quad \mathbf{p}_{\mathrm{adv}} = \sigma\big(\mathcal{F}_\theta(\mathbf{x}_{\mathrm{adv}})\big), \tag{1}$$

where $\sigma$ denotes softmax. Finally, the forget loss is defined by:

$$\mathcal{L}_f(\mathcal{D}_f, \mathbf{p}_{\mathrm{adv}}) = \frac{1}{N_f} \sum_{j=1}^{N_f} \mathcal{L}_{\mathrm{CE}}(x_f^j, p_{\mathrm{adv}}^j), \tag{2}$$

where $\mathcal{L}_{\mathrm{CE}}$ is cross-entropy, and $N_f$ denotes the number of samples. The overall procedure is depicted in the Adversarial Probability Module (APM) in Fig. 2(a).

## 4.2 Pre-trained Generator with OOD Data

Due to our assumption that $\mathcal{D}_r$ is inaccessible, it is essential to practically obtain additional data that can facilitate the extraction of knowledge from $\mathcal{F}_\theta$. Thus, to replace $\mathcal{D}_r$ while obtaining high-quality images, we use a generator trained on an OOD dataset as shown in Fig. 2(b). This strategy offers remarkable flexibility: One may employ readily available open-source datasets or pre-train a generator well in advance of any unlearning requirement.

Previous studies [21, 38, 39] have shown that knowledge distillation (KD) can be performed using OOD data to address similar assumptions that are inaccessible to training data. Fang et al. [21] demonstrated effective KD using OOD data with GANs. Motivated by this, RUAGO employs a generator $G_\psi$, trained on an OOD data to replace the $\mathcal{D}_r$ and preserve model utility.

This strategy provides the advantage of using an open-source pre-trained generator or training the generator with an open-source dataset without $\mathcal{D}_r$. Hence, our approach can always respond to rapid unlearning requests. In RUAGO, $G_\psi$ does not condition on label space $\mathcal{Y}$, which means it cannot assign labels to the generated images. To address this issue, RUAGO calculates the KL divergence between the logits $\mathbf{z}_o$ and $\mathbf{z}_u$, which are obtained by passing the generated images $\tilde{\mathbf{x}}$ from $G_\psi$ through $\mathcal{F}_\theta$ and $\mathcal{F}_\theta^U$, respectively, as shown by the following equation:

$$\mathcal{L}_r(\tilde{\mathbf{x}}) = \frac{1}{N_g} \sum_{k=1}^{N_g} \mathcal{D}_{KL}\left(z_o^k/\tau \parallel z_u^k/\tau\right), \tag{3}$$

where $N_g$ is the number of generated images, $\mathcal{D}_{KL}$ is the KL divergence function, and $\tau$ is a temperature.

## 4.3 Sample Difficulty-Driven Distillation

To effectively transfer information from $\mathcal{F}_\theta$ to $\mathcal{F}_\theta^U$, we design the loss in Eq. (3). However, since $\tilde{\mathbf{x}}$ is generated from the OOD dataset distribution, Eq. (3) may not fully capture comprehensive information. To enhance the distillation process using samples generated by $G_\psi$, RUAGO employs an easy-to-hard training strategy inspired by curriculum learning (CL) [27, 40–44].

**Theorem 1** (Generalization Bound under Sample Difficulty Scheduling). *Let $\mathcal{H}_1 \subseteq \mathcal{H}_2 \subseteq \cdots \subseteq \mathcal{H}_T$ be a nested sequence of hypothesis spaces with VC-dimensions $d_t = \mathrm{VCdim}(\mathcal{H}_t)$ in each curriculum stage t, and define the true risk by $R(\mathcal{F}) = \mathbb{E}_{(x,y)\sim\mathcal{D}}\big[\ell\big(\mathcal{F}(x),y\big)\big]$, and the empirical risk by $\hat{R}_n(\mathcal{F}) = \frac{1}{n}\sum_{i=1}^n \ell\big(\mathcal{F}(x_i),y_i\big)$. Let the original (teacher) model be $\mathcal{F}_\theta \in \mathcal{H}_1$, and let the unlearned (student) model be $\mathcal{F}_\theta^U \in \mathcal{H}_T$. Consider the optimal retrained model $\mathcal{F}_{\theta^*} \in \mathcal{H}_T$ such that $R(\mathcal{F}_{\theta^*}) = \min_{\mathcal{F}\in\mathcal{H}_T} R(\mathcal{F})$. Then, drawing n i.i.d. samples,*

$$R(\mathcal{F}_\theta^U) - R(\mathcal{F}_{\theta^*}) \;\leq\; O\Big(\sum_{t=1}^T \sqrt{\tfrac{d_t \log n}{n}}\Big) \;+\; \epsilon.$$

See Appendix D for the full proof. Theorem 1 provides a theoretical guarantee that our easy-to-hard sample schedule preserves model utility while controlling generalization error. This theorem builds on the principle of Structural Risk Minimization, where the model's effective capacity is gradually increased. In the initial phase, training focuses on "easy" samples, which corresponds to learning within a hypothesis space $\mathcal{H}_t$ with a small VC-dimension $d_t$. As progressively more challenging samples are introduced, the model's effective capacity grows, but it updates from a stable, near-optimal state achieved in the previous stage. To theoretically ground this approach, we adapt a line of VC-based analysis from the related field of DFKD [45], which allows us to ensure robust generalization even with noisy OOD samples.

In RUAGO, we quantify each sample's difficulty via its loss and adjust it dynamically:

$$\hat{\mathcal{L}}_r(\tilde{\mathbf{x}}) = \frac{1}{N_g}\sum_{k=1}^{N_g} w_k\,\ell_k, \quad w_k = \frac{1+\exp(-1/\lambda)}{1+\exp\big(\ell_k - 1/\lambda\big)}, \tag{4}$$

where $\ell_k$ is the loss of the $k$-th generated sample (see Eq. (3)), and $\lambda > 0$ is a temperature hyperparameter controlling the softness of the difficulty weighting. By modulating weight $w_k$ from easy to hard, RUAGO achieves faster, more stable convergence. This loss-based difficulty scheduling serves as a practical implementation of the curriculum described in our theoretical analysis. We begin with easy (low-loss) samples to keep the empirical risk small, preventing divergence before gradually incorporating harder samples. The Sample Difficulty Module (SDM) in Fig. 2(a) illustrates this process. To the best of our knowledge, RUAGO is the first method to leverage sample difficulty scheduling during unlearning, directly addressing OOD challenges in knowledge distillation to boost generalization and model utility.

## 4.4 Inversion-based Generator Alignment

Inspired by model inversion techniques from DFKD [21, 33, 34, 45–48], We insert a brief generator refinement step into our pipeline to ensure that the synthetic samples used in unlearning closely mirror the teacher model's implicit data distribution, thereby promoting a more stable knowledge transfer process. This step optimizes the generator so that its outputs 1) elicit high-confidence predictions from the teacher, 2) expose challenging hard regions of the decision boundary for the student, and 3) match the teacher's internal feature-map statistics. Figure 2(c) represents this procedure.

Concretely, let $\tilde{\mathbf{x}} = G(z)$ be a generated sample. We minimize the following inversion loss:

$$\mathcal{L}_{\mathrm{inv}}(\tilde{\mathbf{x}}) = \gamma_{\mathrm{cls}}\,\mathcal{L}_{\mathrm{CE}}\big(\mathcal{F}_\theta(\tilde{\mathbf{x}}), \arg\max \mathcal{F}_\theta(\tilde{\mathbf{x}})\big)$$

$$- \gamma_{\mathrm{adv}}\,\mathrm{KL}\big(\mathcal{F}_\theta(\tilde{\mathbf{x}}) \,\|\, \mathcal{F}_\theta^U(\tilde{\mathbf{x}})\big)$$

$$+ \gamma_{\mathrm{bn}}\sum_l \Big\|\mu_l - \mu_l^{\mathrm{BN}}\Big\|_2 + \Big\|\sigma_l - \sigma_l^{\mathrm{BN}}\Big\|_2. \tag{5}$$

In particular, the cross-entropy term $\mathcal{L}_{\mathrm{CE}}$ compels the generator to emit samples that the teacher model classifies with high confidence. The negative KL divergence term, by contrast, encourages

the synthesis of challenging hard examples that broaden the output discrepancy between teacher and student. Finally, the statistic-matching term enforces agreement between the batch-wise feature statistics $(\mu_l, \sigma_l)$ and the teacher's stored BatchNorm running statistics $(\mu_l^{\mathrm{BN}}, \sigma_l^{\mathrm{BN}})$. This batch normalization regularizer, primarily used to speed convergence, is only applicable to architectures that include such layers. For models without them, such as Vision Transformers, this term is simply deactivated by setting $\gamma_{\mathrm{bn}} = 0$, with our method still demonstrating strong performance. We adopt this inversion formulation (Deep Inversion [33]) as a drop-in module for high-fidelity sample synthesis. Note that after completing the unlearning process, we safely discard the refined generator to eliminate any potential privacy risks.

## 4.5 Retain-Free Unlearning

We now introduce final unlearning loss of RUAGO, which combines Eq. (2) for forgetting objective and Eq. (4) for retaining objective, formulated as follows:

$$\mathcal{L}_{\mathrm{u}}(\mathcal{D}_f, \mathbf{p}_{\mathrm{adv}}, \tilde{\mathbf{x}}) = \gamma_1 \cdot \mathcal{L}_f(\mathcal{D}_f, \mathbf{p}_{\mathrm{adv}}) + \gamma_2 \cdot \hat{\mathcal{L}}_r(\tilde{\mathbf{x}}), \tag{6}$$

where $\gamma_1$ and $\gamma_2$ are hyper-parameters that control the trade-off between the forgetting and retaining components. Algorithm 1 provides a detailed overview of our method's operational procedure. In each epoch, our approach first performs model inversion. Once the model inversion is completed, it generates the adversarial probability $\mathbf{p}_{\mathrm{adv}}$ and the synthesized image $\tilde{\mathbf{x}}$. These generated components are then used to compute the unlearning loss $\mathcal{L}_{\mathrm{u}}$. The calculated loss is subsequently employed to update the model parameters. This entire process is repeated over multiple epochs. RUAGO enables rapid and effective unlearning without requiring the $\mathcal{D}_r$. Our method suits scenarios where privacy concerns, resource limitations, or expired access rights restrict data storage or access.

---

**Algorithm 1** RUAGO

---

**Input**: $\mathcal{F}_\theta, \mathcal{D}_f, G_\psi$
**Parameters**:
 $\eta, E$ : LR & epochs for unlearning
$\eta_g, E_g$ : LR & epochs for generator alignment
 1: $\mathcal{F}_\theta^U \leftarrow \mathcal{F}_\theta$
 2: **for** $e = 1$ **to** $E$ **do**
 3:   **for** $g = 1$ **to** $E_g$ **do**
 4:     Generate $\tilde{\mathbf{x}}$ and calculate $\mathcal{L}_{\mathrm{inv}}(\tilde{\mathbf{x}})$
 5:     $\psi \leftarrow \psi - \eta_g \cdot \nabla_\psi \mathcal{L}_{\mathrm{inv}}(\tilde{\mathbf{x}})$
 6:   **end for**
 7:   Generate $\mathbf{p}_{\mathrm{adv}}$ and updated $\tilde{\mathbf{x}}$
 8:   Calculate $\mathcal{L}_{\mathrm{u}}(\mathcal{D}_f, \mathbf{p}_{\mathrm{adv}}, \tilde{\mathbf{x}})$
 9:   $\theta \leftarrow \theta - \eta \cdot \nabla_\theta \mathcal{L}_{\mathrm{u}}(\tilde{\mathbf{x}})$
10: **end for**
11: **return** $\mathcal{F}_\theta^U$

---

## 5 Experimental Results

### 5.1 Datasets, Models and Unlearning Setups

We conduct our experiments using the CIFAR-10, CIFAR-100 [49], TinyImageNet [50] and VG-GFace2 [51] datasets. For each dataset, we employ deep learning architectures including VGG16 [52], ResNet18 [53] and Vision Transformer (ViT) [54]. In addition, we employed the COCO dataset [55] as an out-of-distribution resource for training our generative model. We randomly designate 10% of the entire training dataset as $\mathcal{D}_f$, focusing on evaluating instance-wise forgetting. Further experimental details and additional results, including those for CIFAR-100, TinyImageNet, and the deletion of 50% of $\mathcal{D}$, are provided in the Appendix due to space constraints.

### 5.2 Baselines and Evaluation Metrics

In our experiments, we evaluated our method against the Original and Retrain models, as well as six unlearning baselines, including three retain-based methods, Bad-T [17], SCRUB [13], and SalUn [25], and three retain-free methods, Boundary Shrink (BS) [19], SSD [14], and SCAR [16].

To evaluate the performance, we utilize 6 different metrics: 1) RA: accuracy on $\mathcal{D}_r$; 2) UA: accuracy on $\mathcal{D}_f$; 3) TA: accuracy on $\mathcal{D}_t$; 4) AVG: measures the mean of the absolute differences between each method's RA, UA, and TA values and those of the Retrain model; 5) Membership Inference Attack (MIA) [56]: a canonical privacy metric for evaluation unlearning models [13, 14, 16, 17, 19, 25, 57]. The maximized or minimized MIA score could lead to the Streisand effect, unintentionally providing information to attackers. Thus, an MIA value near to the Retrain model is ideal; 6) Running-Time Efficiency (RTE): measure the time efficiency of each method in seconds.

Table 1: Performance of RUAGO and baselines on CIFAR-10 and VGGFace2, reported as mean $\pm$ std, with AVG indicating the accuracy gap between unlearned and retrained models. The "$\mathcal{D}_r$-free" columns (✓ / ✗) marks retain-free methods. **Blue** and **red** highlight the best results for retain-based and retain-free methods, respectively.

(a) Results on CIFAR-10.

| | $\mathcal{D}_r$ free | VGG16 | | | | ResNet18 | | | | ViT | | | |
|---|---|---|---|---|---|---|---|---|---|---|---|---|---|
| | | RA | UA | TA | AVG | RA | UA | TA | AVG | RA | UA | TA | AVG |
| Original | - | 100.00±0.00 | 100.00±0.00 | 93.33±0.35 | - | 99.99±0.00 | 100.00±0.00 | 86.54±0.23 | - | 99.85±0.01 | 99.84±0.05 | 98.97±0.07 | - |
| Retrain | ✗ | 100.00±0.00 | 93.25±0.19 | 92.98±0.19 | 0 | 99.99±0.00 | 86.73±0.58 | 85.96±0.14 | 0 | 99.85±0.01 | 99.03±0.17 | 98.93±0.03 | 0 |
| Bad-T | ✗ | 100.00±0.00 | 93.99±0.55 | 91.88±0.13 | 0.61 | 100.00±0.00 | 85.43±1.96 | 84.59±0.28 | 0.89 | 99.82±0.02 | 99.48±0.12 | 98.82±0.06 | **0.2** |
| SCRUB | ✗ | 99.74±0.32 | 92.04±1.18 | 90.57±0.78 | 1.30 | 100.00±0.00 | 86.87±0.31 | 86.11±0.32 | **0.1** | 99.97±0.00 | 99.86±0.07 | 99.10±0.03 | 0.37 |
| SalUn | ✗ | 100.00±0.00 | 93.68±0.42 | 91.69±0.08 | **0.57** | 100.00±0.00 | 85.82±0.80 | 83.55±0.14 | 1.11 | 99.93±0.01 | 98.13±0.52 | 98.57±0.14 | 0.45 |
| BS | ✓ | 80.49±1.02 | 74.63±0.99 | 73.90±0.63 | 19.07 | 80.20±0.53 | 69.40±1.27 | 69.27±0.36 | 17.93 | 57.27±0.82 | 56.26±0.91 | 56.44±0.81 | 42.62 |
| SSD | ✓ | 74.05±37.58 | 74.17±37.62 | 67.36±33.79 | 23.55 | 79.87±37.95 | 80.06±37.79 | 69.60±32.21 | 14.38 | 83.95±35.53 | 83.91±35.63 | 83.36±35.15 | 15.53 |
| SCAR | ✓ | 93.03±2.24 | 92.90±1.94 | 83.64±1.90 | 5.56 | 88.61±1.27 | 88.19±1.20 | 74.33±0.79 | 8.16 | 98.97±0.41 | 99.08±0.33 | 97.88±0.49 | 0.66 |
| **RUAGO** | ✓ | 99.94±0.02 | 93.38±0.39 | 92.32±0.29 | **0.28** | 99.34±0.15 | 84.02±1.00 | 84.36±0.26 | **1.65** | 99.81±0.01 | 98.95±0.16 | 98.95±0.05 | **0.05** |

(b) Results on VGGFace2.

| | $\mathcal{D}_r$ free | VGG16 | | | | ResNet18 | | | | ViT | | | |
|---|---|---|---|---|---|---|---|---|---|---|---|---|---|
| | | RA | UA | TA | AVG | RA | UA | TA | AVG | RA | UA | TA | AVG |
| Original | - | 98.13±0.15 | 98.59±0.20 | 95.96±0.10 | - | 99.88±0.02 | 99.88±0.02 | 97.52±0.07 | - | 99.24±0.06 | 99.25±0.15 | 96.79±0.18 | - |
| Retrain | ✗ | 98.13±0.15 | 94.46±0.35 | 94.91±0.15 | 0 | 99.88±0.04 | 96.63±0.39 | 97.20±0.18 | 0 | 99.21±0.03 | 96.23±0.32 | 96.55±0.12 | 0 |
| Bad-T | ✗ | 98.45±0.02 | 95.85±1.23 | 95.62±0.19 | **0.81** | 99.84±0.01 | 98.39±1.81 | 96.71±0.11 | **0.76** | 99.15±0.03 | 98.42±0.10 | 96.75±0.08 | **0.82** |
| SCRUB | ✗ | 100.00±0.00 | 98.42±0.05 | 96.92±0.06 | 2.61 | 100.00±0.00 | 98.32±0.79 | 97.80±0.08 | 0.8 | 99.76±0.10 | 99.27±0.15 | 96.96±0.11 | 1.33 |
| SalUn | ✗ | 99.61±0.07 | 93.66±1.41 | 95.17±0.25 | 0.85 | 99.99±0.00 | 98.71±0.27 | 96.97±0.14 | 0.81 | 99.26±0.19 | 93.03±1.28 | 95.26±0.44 | 1.51 |
| BS | ✓ | 95.13±0.12 | 95.13±0.19 | 91.86±0.19 | 2.24 | 96.53±0.12 | 96.61±0.42 | 91.99±0.15 | 2.86 | 89.45±0.69 | 89.70±0.66 | 86.52±0.67 | 8.77 |
| SSD | ✓ | 88.23±5.39 | 88.29±5.83 | 85.32±5.15 | 8.56 | 95.05±2.95 | 95.22±2.86 | 91.32±3.30 | 4.04 | 97.49±1.07 | 97.43±1.24 | 94.87±1.09 | 1.54 |
| SCAR | ✓ | 95.24±0.16 | 95.40±0.16 | 91.79±0.28 | 2.32 | 96.84±0.64 | 96.97±0.70 | 92.86±0.89 | 2.57 | 96.53±0.47 | 96.62±0.29 | 93.71±0.37 | 1.97 |
| **RUAGO** | ✓ | 97.44±0.06 | 95.44±0.26 | 94.88±0.17 | **0.57** | 99.32±0.05 | 96.64±0.22 | 96.41±0.15 | **0.45** | 98.91±0.02 | 96.94±0.20 | 96.28±0.09 | **0.42** |

## 5.3 Results

**Accuracy Performance.** Table 1 presents unlearning accuracy on two datasets for the original model, the retrained model, six baselines, and RUAGO. Table 1(a) presents the accuracy results for CIFAR-10 dataset, indicating that retain-free baselines are largely ineffective at unlearning, regardless of the model type. In the worst case, an AVG value of 42.62 is observed. The similar RA and UA values across the three methods indicate that a general performance degradation has occurred rather than erasing $\mathcal{D}_f$. In contrast, RUAGO demonstrates remarkable performance. The AVG values indicate that RUAGO outperforms the other three methods, with RA, UA, and TA closely aligning with those of the Retrain model. Although, Bad-T, SCRUB, and SalUn are slightly more effective than RUAGO for the ResNet18 model, our method still demonstrates comparable performance. Notably, RUAGO outperforms these methods for VGG16 and ViT, with AVG values of 0.28 and 0.05, respectively, indicating the closest match to the Retrain model.

Table 1(b) presents the results for the VGGFace2 dataset. Retain-free methods, such as Boundary Shrink, SSD, and SCAR, exhibit significant performance degradation, similar to the CIFAR-10 results. In contrast, our approach outperforms all other methods, including retain-based methods. This result indicates that even when $G_\psi$ is trained on an OOD dataset (COCO) that is significantly different from $\mathcal{D}$ (VGGFace2), it still supports the unlearning process effectively. In conclusion, although unlearning without $\mathcal{D}_r$ is challenging, RUAGO succeeds admirably, comparable to or outperforming other baselines.

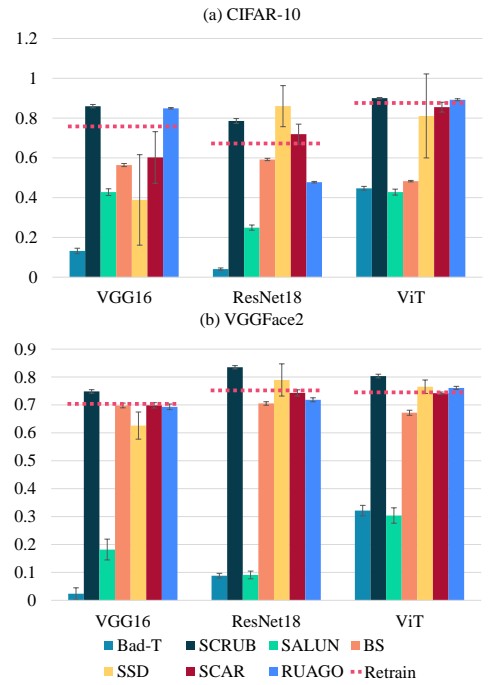

Figure 3: MIA results for each unlearning method. The red dotted line represents the Retrain model; it is best to be closer to this line.

Table 2: Running Time Efficiency (RTE) results measured in seconds.

|  | CIFAR-10 | | | VGGFace2 | | |
| --- | --- | --- | --- | --- | --- | --- |
|  | VGG16 | ResNet18 | ViT | VGG16 | ResNet18 | ViT |
| Retrain | 1,758 | 1,991 | 4,862 | 5,071 | 4,735 | 5,790 |
| Bad-T | 181 | 215 | 2,105 | 5,403 | 1,200 | 2,418 |
| SCRUB | 172 | 203 | 1,752 | 4,844 | 1,073 | 1,993 |
| SalUn | 161 | 184 | 1,502 | 4,005 | 1,146 | 1,755 |
| BS | 55 | 83 | 514 | 1,599 | 446 | 153 |
| SSD | 13 | 15 | 581 | 299 | 59 | 685 |
| SCAR | 26 | 35 | 333 | 133 | 68 | 466 |
| **RUAGO** | 433 | 473 | 532 | 892 | 395 | 625 |

Table 3: Results of the ablations studies for the VGG16 model on CIFAR-10.

|  | RA | UA | TA | MIA | AVG |
| --- | --- | --- | --- | --- | --- |
| **RUAGO** | 99.94 | 93.38 | 92.32 | 0.85 | 0.282 |
| Hard labels | 96.61 | 93.93 | 87.62 | 0.51 | 3.146 |
| Diff. OOD | 99.93 | 93.55 | 92.20 | 0.85 | 0.384 |
| w/ Init $G_\psi$ | 72.55 | 72.45 | 68.22 | 0.60 | 24.338 |
| w/o MI | 99.95 | 93.48 | 92.38 | 0.85 | 0.297 |
| w/o SD | 99.47 | 88.93 | 91.44 | 0.86 | 2.133 |

**MIA Score.** To evaluate privacy leakage, we analyze the MIA scores of all methods. The Bad-T method, which shows excellent accuracy performance, records the worst MIA score. This suggests that the Bad-T approach does not guarantee privacy protection. In contrast, the MIA values of SCAR show a minor discrepancy from the Retrain model, but it shows poor accuracy, as mentioned above. As a result, the evaluation of effective unlearning requires a comprehensive set metrics. Our method, which performs accurately, also demonstrates strong privacy protection. As shown in Fig. 3, our approach achieves MIA values comparable to those of the Retrain model in all experiments except for ResNet18 trained on CIFAR-10. Particularly for ViT on both datasets, the MIA scores are almost identical to the Retrain model, with only about a 1% difference. This indicates that RUAGO exhibits excellent unlearning performance in accuracy and protecting privacy.

**Running-Time Efficiency.** Efficient unlearning requires latency that is lower than that of a full retraining procedure. Table 2 reports each unlearning method's running-time efficiency (RTE), measured in seconds. Notably, this metric omits all preparatory computations, such as the Fisher information matrix estimation in SSD, the mean and covariance computations in SCAR, and the out-of-distribution generator training in our approach, and concentrates solely on post-request execution time. Although these offline steps can be time-consuming, they may be precomputed without affecting the immediate unlearning latency. Methods that depend on the retain dataset $\mathcal{D}_r$ (Bad-T, SCRUB, and SalUn) incur substantial RTE costs, especially when applied to large architectures like ViT or high-resolution datasets such as VGGFace2. In contrast, techniques that do not use $\mathcal{D}_r$ (BS, SSD, and SCAR) achieve lower RTE values but do not deliver effective unlearning performance, as noted above. Our method, however, strikes an ideal balance: it matches or exceeds the efficiency of retain-free baselines (providing up to a 12× speed-up over full retraining) without sacrificing unlearning efficacy. These results underscore the practical advantage of our framework for real-time unlearning, even in the most challenging scenarios where the retain data is unavailable.

## 5.4 Ablation Studies

In this section, we present ablation studies demonstrating the importance of various design choices in RUAGO. We summarize the main findings here, with additional results and analyses provided in Appendix G.

**Component Analysis.** We first analyze the impact of each component using the VGG16 model on CIFAR-10. As shown in Table 3, each component of RUAGO plays a crucial role. First, unlearning with hard labels instead of our soft targets significantly reduces test accuracy (TA) and yields membership inference attack (MIA) scores that diverge from the retrained model, indicating lower utility and potential privacy risks. Second, employing generators trained on different OOD datasets, such as TinyImageNet, consistently ensures robust unlearning, highlighting RUAGO's versatility. Third, using a randomly initialized generator leads to markedly poor performance, underscoring the necessity of a pre-trained OOD generator. Fourth, while RUAGO is effective without the model inversion technique, its inclusion further improves unlearning outcomes. Lastly, omitting the sample difficulty scheduling from the distillation process critically harms model utility. These findings collectively validate our design choices in achieving efficient and robust unlearning.

**Hyperparameter Sensitivity Analysis.** To address concerns about the hyperparameter space, we conducted a sensitivity analysis on the loss weights ($\gamma_{adv}, \gamma_{bn}, \gamma_{cls}, \gamma_1, \gamma_2$) for the VGG16 model on CIFAR-10, varying one while keeping others fixed. The detailed results are presented in the Appendix (Tables 9 and 10). Our findings show that the performance of RUAGO is not overly sensitive to its hyperparameters. Key metrics remained stable across a wide range of values, demonstrating the robustness of our method. This leads to a practical and efficient tuning guideline. In our experiments,

Table 4: Effect of the number of alignment epochs ($E_g$) across different models on VGGFace2. Metrics remain stable or slightly improve as $E_g$ increases, demonstrating that the alignment process is safe and does not re-inject forgotten information across diverse architectures.

| $E_g$ | VGG16 | | | ResNet18 | | | ViT | | |
|---|---|---|---|---|---|---|---|---|---|
| | RA | UA | TA | RA | UA | TA | RA | UA | TA |
| 10 | 97.44 | 95.44 | 94.88 | 99.32 | 96.64 | 96.41 | 98.91 | 96.94 | 96.28 |
| 30 | 97.53 | 95.43 | 94.90 | 99.43 | 96.46 | 96.51 | 98.92 | 96.88 | 96.31 |
| 50 | 97.57 | 95.43 | 94.95 | 99.47 | 96.43 | 96.57 | 98.92 | 96.89 | 96.32 |
| 70 | 97.59 | 95.54 | 94.98 | 99.47 | 96.41 | 96.48 | 98.92 | 96.89 | 96.31 |
| 100 | 97.62 | 95.57 | 95.01 | 99.49 | 96.38 | 96.60 | 98.92 | 96.87 | 96.32 |

a simple one-dimensional sweep for $\gamma_1$ after fixing other weights was sufficient to find a near-optimal configuration. This demonstrates that RUAGO can achieve strong performance without extensive, multi-dimensional hyperparameter tuning.

**Analysis of Alignment and Re-injection Risk.** A critical consideration is whether the generator alignment step could inadvertently re-inject forgotten information. We mitigate this risk through safeguards like the forget loss $\mathcal{L}_f$. In the class-wise unlearning scenario, we employ an additional safeguard of output filtering. To empirically validate the safety of this component, we analyzed the effect of alignment strength. As detailed in the Appendix (Table 11), using a high learning rate to induce overly aggressive alignment leads to a general performance collapse rather than selective re-injection, indicating optimization instability.

Furthermore, we investigated the impact of the number of alignment epochs ($E_g$) across multiple architectures, as shown in Table 4. The results are highly consistent, as increasing the alignment epochs from 10 to 100 maintains robust unlearning performance (stable UA) while slightly improving utility (RA and TA) across VGG16, ResNet18, and ViT. This compellingly demonstrates that the alignment module operates within a safe regime, stabilizing distillation without re-introducing forgotten data, regardless of the model architecture.

## 6 Conclusion

We propose RUAGO, a unified retain-free unlearning framework that prevents over-unlearning issues and preserves the model's utility without access to retain data. We aim to tackle the performance drop common in retain-free scenarios, especially in instance-wise unlearning. Our method generates adversarial probabilities to prevent forget samples from being pushed beyond decision boundaries, thereby mitigating over-unlearning. Simultaneously, to compensate for the absence of retain data, we leverage a generator pretrained on OOD data to distill knowledge from the original model using synthetic samples. To address potential instability caused by synthetic samples and to enhance generalization during distillation, we incorporate a sample scheduling strategy informed by VC theory, and apply inversion-based alignment to adjust the generator's outputs toward the original data distribution. These components form a cohesive framework that achieves effective and practical unlearning in retain-free scenarios. Our extensive experimental results show that RUAGO outperforms existing retain-free methods, and matches or exceeds retain-based approaches, confirming its practicality for real-world unlearning scenarios without the retain data.

## Acknowledgement

The authors would thank anonymous reviewers. This work was partly supported by Institute for Information & communication Technology Planning & evaluation (IITP) grants funded by the Korean government MSIT: (RS-2022-II221199, RS-2022-II220688, RS-2019-II190421, RS-2023-00230337, RS-2024-00437849, RS-2021-II212068, and RS-2025-02263841). Also, this work was supported by the Cyber Investigation Support Technology Development Program (No.RS-2025-02304983) of the Korea Institute of Police Technology (KIPoT), funded by the Korean National Police Agency. Lastly, this work was supported by the National Research Foundation of Korea (NRF) grant funded by the Korea government (MSIT) (No. RS-2024-00356293).

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

## NeurIPS Paper Checklist

The checklist is designed to encourage best practices for responsible machine learning research, addressing issues of reproducibility, transparency, research ethics, and societal impact. Do not remove the checklist: **The papers not including the checklist will be desk rejected.** The checklist should follow the references and follow the (optional) supplemental material. The checklist does NOT count towards the page limit.

Please read the checklist guidelines carefully for information on how to answer these questions. For each question in the checklist:

- You should answer [Yes] , [No] , or [NA] .
- [NA] means either that the question is Not Applicable for that particular paper or the relevant information is Not Available.
- Please provide a short (1–2 sentence) justification right after your answer (even for NA).

**The checklist answers are an integral part of your paper submission.** They are visible to the reviewers, area chairs, senior area chairs, and ethics reviewers. You will be asked to also include it (after eventual revisions) with the final version of your paper, and its final version will be published with the paper.

The reviewers of your paper will be asked to use the checklist as one of the factors in their evaluation. While "[Yes] " is generally preferable to "[No] ", it is perfectly acceptable to answer "[No] " provided a proper justification is given (e.g., "error bars are not reported because it would be too computationally expensive" or "we were unable to find the license for the dataset we used"). In general, answering "[No] " or "[NA] " is not grounds for rejection. While the questions are phrased in a binary way, we acknowledge that the true answer is often more nuanced, so please just use your best judgment and write a justification to elaborate. All supporting evidence can appear either in the main paper or the supplemental material, provided in appendix. If you answer [Yes] to a question, in the justification please point to the section(s) where related material for the question can be found.

IMPORTANT, please:

- **Delete this instruction block, but keep the section heading "NeurIPS Paper Checklist",**
- **Keep the checklist subsection headings, questions/answers and guidelines below.**
- **Do not modify the questions and only use the provided macros for your answers**.


## A   Datasets

### A.1   CIFAR-10 and CIFAR-100

The CIFAR-10 and CIFAR-100 datasets [49] are widely used benchmarks for image classification. Each dataset contains 60,000 color images at a resolution of 32×32 pixels. CIFAR-10 has 10 classes, such as airplanes, automobiles, and animals, while CIFAR-100 expands this to 100 fine-grained classes. Both datasets are split into a training set of 50,000 images and a test set of 10,000 images. These datasets are valuable in the computer vision area for evaluating the performance of classification models due to their balanced class distribution and moderate complexity.

### A.2   TinyImageNet

TinyImageNet dataset [50] is an image classification dataset commonly used in deep learning research. It is a smaller version of the original ImageNet dataset [58]. The dataset comprises 200 classes, with each class containing 500 training images and 50 validation images. Each image has a resolution of 64×64 pixels, making it smaller than the original ImageNet dataset. The dataset includes 100,000 training images, 10,000 validation images, and 10,000 test images. We utilize the train set and validation set for experiments.

### A.3   VGGFace2

The VGGFace2 dataset [52] is a large-scale collection intended for face recognition applications. This dataset comprises facial data and is related to privacy preservation tasks. The dataset's high similarity among classes makes it essential for evaluating the effectiveness of unlearning methods in practical applications involving facial data. The dataset comprises various facial images that differ in identity, pose, illumination, background, and expression. The dataset comprises over 3.31 million images sourced from more than 9,000 individuals. To conduct our unlearning task, we randomly selected 100 individuals from a training dataset and resized them into 224×224 resolution.

### A.4   Stanford Cars

The Stanford Cars [59] dataset is a fine-grained image classification benchmark for car recognition. It consists of 196 classes, categorized by the manufacturer, model, and year of the vehicle. The dataset includes 8,144 training images and 8,041 test images. Many cars from the same manufacturer or similar models and years share very similar designs. Thanks to these characteristics, the Stanford Cars dataset is useful for evaluating how effectively unlearning methods can remove fine-grained visual details.

## B   Metrics

### B.1   Accuracy

In order to assess a classifier's performance, accuracy is frequently utilized. It measures the percentage of samples for which the true classes can be predicted with maximum certainty. Accuracy of a model $\mathcal{F}$ tested on a dataset of $N$ samples $\{(x^1, y^1), ..., (x^N, y^N)\}$ is formulated as follows:

$$\text{ACC} = 100 \cdot \frac{\sum_{i=1}^{N} \delta(\sigma(\mathcal{F}(x^i)), y^i)}{N},$$

where $\delta(\cdot, \cdot)$ is the Kronecker delta function.

## B.2 Membership Inference Attack Score

Membership inference attack (MIA) [56] is a metric used to determine whether a specific data point was used during the model training. We employ MIA to verify the effectiveness of unlearning. To compute the MIA score, we pass the retain set ($\mathcal{D}_r$) and the test set ($\mathcal{D}_t$) through the target model to obtain probabilities for each data point:

$$\mathbf{p}_r = \mathcal{F}_\theta^U(\mathcal{D}_r), \ \ \mathbf{p}_t = \mathcal{F}_\theta^U(\mathcal{D}_t)$$

After that, We assign label 1 to the probabilities from $\mathcal{D}_r$ and label 0 to those from $\mathcal{D}_t$, and train a shallow model using these labeled data points:

$$\mathcal{F}_{\text{shallow}}(\{(\mathbf{p}_r, \mathbf{1}), (\mathbf{p}_f, \mathbf{0})\})$$

In our case, we used the LogisticRegression model from Scikit-learn [60]. Following this, we pass the forget set ($\mathcal{D}_f$) through the target model to compute probabilities for each data point and use the shallow model to predict labels:

$$\mathbf{p}_f = \mathcal{F}_\theta^U(\mathcal{D}_f), \ \ \hat{y}_f = \mathcal{F}_{\text{shallow}}(\mathbf{p}_f)$$

The average of these predicted values is used as the MIA score:

$$\text{MIA score} = \frac{1}{N_f} \sum_{j=1}^{N_f} \hat{y}_f^j$$

## B.3 Running Time Efficiency

To measure the Running Time Efficiency (RTE), we record the time taken for the entire unlearning process, from the execution of the unlearning algorithm to its completion, expressed in seconds. This measurement excludes any preparations, such as data loading, preprocessing, or any metadata computation that may need to be conducted in advance. These steps are not included in the RTE metric to ensure a fair comparison that focuses solely on the efficiency of the unlearning methodology itself.

## C Implementation and Hyperparameter Settings

We set the hyperparameters for RUAGO as follows. The weights $\gamma_{\text{cls}}$, $\gamma_{\text{adv}}$, and $\gamma_{\text{bn}}$ were selected within the range $[0, 5]$. The coefficient $\gamma_1$ was varied within $[0.01, 1.50]$, while $\gamma_2$ was fixed at 0.01. The learning rate was adjusted within $[5 \times 10^{-6}, 5 \times 10^{-4}]$, and training was performed for 1 to 50 epochs. We used the SGD optimizer for training the original model and the Adam optimizer [61] for unlearning baselines. For adversarial attacks, we applied the PGD attack [36]. All experiments were conducted using five different random seeds on a single NVIDIA RTX A5000 GPU.

## D Proof

**Theorem 1** (Generalization Bound under Sample Difficulty Scheduling). *Let $\mathcal{H}_1 \subseteq \mathcal{H}_2 \subseteq \cdots \subseteq \mathcal{H}_T$ be a nested sequence of hypothesis spaces with VC-dimensions $d_t = \text{VCdim}(\mathcal{H}_t)$ in each curriculum stage $t$, and define the true risk by $R(\mathcal{F}) = \mathbb{E}_{(x,y) \sim \mathcal{D}}[\ell(\mathcal{F}(x), y)]$, and the empirical risk by $\hat{R}_n(\mathcal{F}) = \frac{1}{n} \sum_{i=1}^n \ell(\mathcal{F}(x_i), y_i)$. Let the original (teacher) model be $\mathcal{F}_\theta \in \mathcal{H}_1$, and let the unlearned (student) model be $\mathcal{F}_\theta^U \in \mathcal{H}_T$. Consider the optimal retrained model $\mathcal{F}_{\theta^*} \in \mathcal{H}_T$ such that $R(\mathcal{F}_{\theta^*}) = \min_{\mathcal{F} \in \mathcal{H}_T} R(\mathcal{F})$. Then, drawing $n$ i.i.d. samples,*

$$R(\mathcal{F}_\theta^U) - R(\mathcal{F}_{\theta^*}) \leq O\left(\sum_{t=1}^T \sqrt{\frac{d_t \log n}{n}}\right) + \epsilon.$$

*Proof.* Fix $\delta > 0$. For each $t = 1, \ldots, T$, define

$$\varepsilon_t = C\sqrt{\frac{d_t\big(\log(n/d_t) + \log(T/\delta)\big)}{n}} = O\Big(\sqrt{\tfrac{d_t \log n}{n}}\Big),$$

where $C > 0$ is a universal constant from the VC uniform-convergence bound, $d_t \geq 1$ is the (finite) VC-dimension of the stage-t hypothesis space, $n \geq 1$ is the sample size, and $\delta \in (0, 1)$, respectively. Hence $\epsilon_t > 0$. By the standard VC uniform-convergence theorem [62–64], for any single hypothesis space $\mathcal{H}_t$,

$$\Pr\left[\sup_{\mathcal{F} \in \mathcal{H}_t} \big|R(\mathcal{F}) - \hat{R}_n(\mathcal{F})\big| \leq \varepsilon_t\right] \geq 1 - \tfrac{\delta}{T}.$$

To ensure this bound holds simultaneously for all $T$ stages, we apply a union bound. Thus, with probability at least $1 - \delta$, we have

$$\big|R(\mathcal{F}) - \hat{R}_n(\mathcal{F})\big| \leq \varepsilon_t \quad \text{for every } \mathcal{F} \in \mathcal{H}_t, \ t = 1, \ldots, T.$$

Let

$$\mathcal{F}_1 = \mathcal{F}_\theta, \quad \mathcal{F}_T = \mathcal{F}_\theta^U, \quad \mathcal{F}^* = \mathcal{F}_{\theta^*},$$

and define

$$\epsilon = R(\mathcal{F}_1) - R(\mathcal{F}^*),$$

which is a fixed constant since $\mathcal{F}_1$ and $\mathcal{F}^*$ are fixed; hence, their error gap depends on the generator. Then

$$R(\mathcal{F}_T) - R(\mathcal{F}^*) = \big[R(\mathcal{F}_T) - R(\mathcal{F}_1)\big] + \big[R(\mathcal{F}_1) - R(\mathcal{F}^*)\big] = \big[R(\mathcal{F}_T) - R(\mathcal{F}_1)\big] + \epsilon.$$

Next, telescope the first term:

$$R(\mathcal{F}_T) - R(\mathcal{F}_1) = \sum_{t=2}^{T}\big[R(\mathcal{F}_t) - R(\mathcal{F}_{t-1})\big].$$

For each $t = 2, \ldots, T$, using $\mathcal{H}_{t-1} \subseteq \mathcal{H}_t$ and the fact that $\mathcal{F}_t$ minimizes the empirical risk over $\mathcal{H}_t$, we have $\hat{R}_n(\mathcal{F}_t) \leq \hat{R}_n(\mathcal{F}_{t-1})$. Thus,

$$\begin{aligned} R(\mathcal{F}_t) - R(\mathcal{F}_{t-1}) &= \big(R(\mathcal{F}_t) - \hat{R}_n(\mathcal{F}_t)\big) + \big(\hat{R}_n(\mathcal{F}_t) - \hat{R}_n(\mathcal{F}_{t-1})\big) + \big(\hat{R}_n(\mathcal{F}_{t-1}) - R(\mathcal{F}_{t-1})\big) \\ &\leq \big|R(\mathcal{F}_t) - \hat{R}_n(\mathcal{F}_t)\big| + \big(\hat{R}_n(\mathcal{F}_t) - \hat{R}_n(\mathcal{F}_{t-1})\big) + \big|\hat{R}_n(\mathcal{F}_{t-1}) - R(\mathcal{F}_{t-1})\big| \\ &\leq \varepsilon_t + 0 + \varepsilon_{t-1} = \varepsilon_t + \varepsilon_{t-1}, \end{aligned}$$

The first line is an identity. The second line follows from the property that $x \leq |x|$. The third line follows because: (1) by the uniform convergence bound, $|R(\mathcal{F}_t) - \hat{R}_n(\mathcal{F}_t)| \leq \epsilon_t$ and $|\hat{R}_n(\mathcal{F}_{t-1}) - R(\mathcal{F}_{t-1})| \leq \epsilon_{t-1}$; and (2) since $\mathcal{F}_t$ minimizes the empirical risk over $\mathcal{H}_t$ and $\mathcal{F}_{t-1} \in \mathcal{H}_t$ (due to the nested spaces), we have $\hat{R}_n(\mathcal{F}_t) \leq \hat{R}_n(\mathcal{F}_{t-1})$, which implies the middle term is non-positive. Summing over $t = 2, \ldots, T$ gives

$$R(\mathcal{F}_T) - R(\mathcal{F}_1) \leq \sum_{t=2}^{T}(\varepsilon_t + \varepsilon_{t-1}) \leq 2\sum_{t=1}^{T}\varepsilon_t.$$

Therefore

$$R(\mathcal{F}_\theta^U) - R(\mathcal{F}_{\theta^*}) \leq 2\sum_{t=1}^{T}\varepsilon_t + \epsilon.$$

Noting that each $\varepsilon_t = O\big(\sqrt{d_t \log n / n}\big)$, we conclude

$$R(\mathcal{F}_\theta^U) - R(\mathcal{F}_{\theta^*}) \leq O\Big(\sum_{t=1}^{T}\sqrt{\tfrac{d_t \log n}{n}}\Big) + \epsilon.$$

$\square$

Table 5: Performance of RUAGO and baselines on CIFAR-100 and TinyImageNet, reported as mean $\pm$ std, with AVG indicating the accuracy gap between unlearned and retrained models. The "$\mathcal{D}_r$-free" columns (✓ / ✗) marks retain-free methods. **Blue** and red highlight the best results for retain-based and retain-free methods, respectively.

(a) Results on CIFAR-100.

| | $\mathcal{D}_r$ free | VGG16 RA | UA | TA | AVG | ResNet18 RA | UA | TA | AVG | ViT RA | UA | TA | AVG |
|---|---|---|---|---|---|---|---|---|---|---|---|---|---|
| Original | - | 99.98±0.00 | 99.98±0.02 | 72.93±0.24 | - | 99.97±0.00 | 99.99±0.02 | 56.55±0.32 | - | 98.43±0.41 | 98.30±0.52 | 92.62±0.28 | - |
| Retrain | ✗ | 99.98±0.00 | 71.92±0.71 | 71.89±0.26 | 0 | 99.97±0.00 | 64.18±20.01 | 55.44±0.41 | 0 | 98.45±0.39 | 92.16±0.53 | 92.51±0.31 | 0 |
| Bad-T | ✗ | 99.97±0.00 | 71.47±0.76 | 70.05±0.10 | 0.77 | 99.91±0.01 | 45.14±2.67 | 51.52±0.35 | 7.67 | 92.93±0.25 | 92.97±0.15 | 86.92±0.17 | 0.5 |
| SCRUB | ✗ | 99.88±0.10 | 75.91±6.45 | 69.89±0.34 | 2.03 | 99.70±0.08 | 55.16±0.58 | 54.34±0.28 | 3.46 | 99.32±0.02 | 97.35±0.64 | 93.03±0.08 | 2.19 |
| SalUn | ✗ | 99.98±0.00 | 67.93±1.00 | 66.63±0.16 | 3.09 | 99.98±0.01 | 46.75±2.18 | 47.61±0.19 | 8.42 | 98.74±0.18 | 96.78±0.20 | 92.47±0.26 | 1.65 |
| BS | ✓ | 80.48±0.76 | 78.40±0.55 | 54.70±0.41 | 14.39 | 61.15±0.58 | 45.95±0.82 | 36.91±0.34 | 25.19 | 97.32±0.04 | 97.26±0.15 | 91.93±0.10 | 2.27 |
| SSD | ✓ | 89.15±10.78 | 89.06±10.79 | 63.27±8.59 | 12.2 | 86.34±13.98 | 86.21±13.75 | 46.34±6.12 | 14.92 | 92.22±2.59 | 91.86±2.81 | 87.50±2.48 | 3.84 |
| SCAR | ✓ | 71.93±1.83 | 71.40±1.95 | 45.22±1.07 | 18.42 | 58.33±0.62 | 57.53±0.39 | 24.06±0.37 | 26.56 | 92.93±0.25 | 92.97±0.15 | 86.92±0.17 | 3.97 |
| **RUAGO** | ✓ | 99.03±0.13 | 67.00±0.98 | 68.70±0.35 | 3.02 | 99.01±0.18 | 58.90±1.48 | 52.42±0.53 | 3.09 | 98.51±1.18 | 96.32±2.46 | 94.82±3.75 | 2.18 |

(b) Results on TinyImageNet.

| | $\mathcal{D}_r$ free | VGG16 RA | UA | TA | AVG | ResNet18 RA | UA | TA | AVG | ViT RA | UA | TA | AVG |
|---|---|---|---|---|---|---|---|---|---|---|---|---|---|
| Original | - | 99.98±0.00 | 99.98±0.02 | 58.97±0.09 | - | 99.98±0.00 | 99.98±0.01 | 45.67±0.32 | - | 96.22±0.04 | 96.28±0.10 | 90.98±0.06 | - |
| Retrain | ✗ | 99.98±0.00 | 58.19±0.44 | 58.09±0.33 | 0 | 99.98±0.00 | 44.63±0.44 | 44.35±0.32 | 0 | 96.21±0.03 | 90.82±0.28 | 91.03±0.24 | 0 |
| Bad-T | ✗ | 99.98±0.00 | 51.59±1.63 | 54.85±0.26 | 3.28 | 98.83±0.16 | 14.68±3.15 | 36.33±0.40 | 13.04 | 96.00±0.02 | 89.10±0.10 | 90.63±0.14 | 0.78 |
| SCRUB | ✗ | 99.98±0.00 | 96.41±1.30 | 58.60±0.22 | 12.91 | 99.98±0.00 | 99.96±0.01 | 46.02±0.07 | 19 | 97.32±1.72 | 95.01±1.61 | 90.15±1.37 | 2.06 |
| SalUn | ✗ | 99.97±0.01 | 9.24±14.42 | 44.12±1.43 | 20.98 | 99.98±0.01 | 10.07±15.03 | 32.51±2.04 | 15.47 | 97.90±0.00 | 94.83±0.08 | 90.67±0.09 | 2.02 |
| BS | ✓ | 62.72±0.71 | 55.25±0.85 | 35.64±0.29 | 20.88 | 73.14±0.72 | 55.19±0.67 | 31.09±0.39 | 16.89 | 96.29±0.02 | 96.26±0.17 | 90.99±0.01 | 1.86 |
| SSD | ✓ | 84.33±13.18 | 84.01±13.06 | 46.02±7.72 | 17.85 | 71.17±25.56 | 70.98±25.68 | 30.97±10.00 | 22.85 | 85.62±10.68 | 85.42±10.25 | 80.93±9.74 | 8.7 |
| SCAR | ✓ | 57.04±0.72 | 57.03±0.99 | 27.23±1.56 | 24.99 | 46.06±0.82 | 45.93±0.36 | 13.49±0.38 | 28.7 | 91.37±0.36 | 91.21±0.20 | 85.53±0.32 | 3.58 |
| **RUAGO** | ✓ | 99.87±0.02 | 55.65±0.62 | 55.02±0.08 | 1.91 | 98.66±0.15 | 41.52±1.10 | 39.48±0.22 | 3.1 | 95.98±0.03 | 92.44±0.23 | 90.56±0.06 | 0.77 |

# E    Additional Experiments

We additionally conducted experiments using the CIFAR-100 and TinyImageNet datasets. The results for CIFAR-100 are summarized in Table 5(a). Consistent with the findings from previous experimental results in Section 5.3, RUAGO demonstrates the most robust performance among retain-free methods. Retain-free baselines, such as those evaluated on VGG16 and ResNet18 models, demonstrate AVG values exceeding 10.00. In contrast, our proposed method achieves an AVG value of approximately 3.00, indicating a significant improvement over the other approaches. Additionally, RUAGO demonstrates unlearning performance comparable to retain-based methods. Notably, for the ResNet18 model, RUAGO achieves an AVG value of 3.09, representing the highest unlearning performance across all baselines. In terms of the MIA score, our method effectively mitigates privacy leakage. As shown in Fig. 4(a), Bad-T (Bad-T), which demonstrated the highest accuracy performance, exhibits a substantial discrepancy in the MIA score compared to the Retrain model. This discrepancy suggests a potential Streisand effect, which could lead to significant privacy leakage. On the other hand, while the SCAR method achieves an MIA score close to that of the Retrain model, it falls short in accuracy performance, as indicated in Table 5(a). Unlike these methods, RUAGO not only achieves superior unlearning performance in accuracy but also shows an MIA score similar to that of the Retrain model, indicating better privacy protection compared to other methods.

The experimental results on TinyImageNet, as shown in Table 5(b), further confirm that our method significantly outperforms retain-free baselines. These findings indicate that retain-free approaches struggle with datasets containing many classes, such as TinyImageNet, which includes 200 classes. In contrast, RUAGO demonstrates remarkable unlearning performance under these challenging conditions. Furthermore, when compared to retain-based methods, our approach consistently achieves superior results. Specifically, for the ResNet18 model, competing methods fail to deliver satisfactory unlearning outcomes, deviating from the desired unlearning objectives. Conversely, RUAGO achieves an AVG value of 3.1, the best among all baselines. Regarding the MIA score, our approach consistently achieves an MIA score relatively close to that of the Retrain model as illustrated in Fig. 4(b), similar to results observed on other datasets. This alleviates concerns about the Streisand effect. Furthermore, our method demonstrates superior accuracy performance, indicating its capacity to achieve a remarkable equilibrium between model efficacy and privacy safeguarding. These results emphasize the effectiveness of RUAGO in the instance-wise unlearning scenario, even in settings without access to $\mathcal{D}_r$.

Table 6: Performance comparison for unlearning 50% of $\mathcal{D}$ on the CIFAR-10 and VGGFace2 datasets.

(a) Results on CIFAR-10

| | $\mathcal{D}_r$ free | VGG16 | | | | ResNet18 | | | | ViT | | | |
|---|---|---|---|---|---|---|---|---|---|---|---|---|---|
| | | RA | UA | TA | AVG | RA | UA | TA | AVG | RA | UA | TA | AVG |
| Original | - | 100.00±0.00 | 100.00±0.00 | 93.33±0.35 | - | 99.99±0.00 | 100.00±0.00 | 86.54±0.23 | - | 99.85±0.01 | 99.84±0.05 | 98.97±0.07 | - |
| Retrain | ✗ | 100.00±0.00 | 90.79±0.18 | 90.19±0.28 | 0 | 100.00±0.00 | 81.55±0.03 | 81.03±0.31 | 0 | 99.84±0.02 | 98.96±0.02 | 98.79±0.05 | 0 |
| Bad-T | ✗ | 99.98±0.02 | 89.83±4.29 | 86.19±2.22 | **1.66** | 99.98±0.00 | 82.32±1.10 | 77.51±0.22 | **1.43** | 99.47±0.07 | 98.88±0.14 | 98.01±0.16 | 0.41 |
| SCRUB | ✗ | 100.00±0.00 | 99.84±0.02 | 92.97±0.09 | 3.94 | 100.00±0.00 | 99.73±0.05 | 86.95±0.05 | 8.03 | 99.90±0.02 | 99.86±0.02 | 99.12±0.01 | 0.43 |
| SalUn | ✗ | 99.92±0.06 | 88.96±1.31 | 87.01±0.41 | 1.70 | 99.26±0.16 | 81.49±1.49 | 75.19±0.81 | 2.22 | 99.73±0.08 | 98.97±0.09 | 98.24±0.11 | **0.22** |
| BS | ✓ | 98.67±0.04 | 98.57±0.14 | 89.96±0.12 | 3.11 | 90.99±0.36 | 90.84±0.37 | 77.02±0.22 | 7.44 | 99.71±0.03 | 99.74±0.02 | 98.89±0.01 | 0.34 |
| SSD | ✓ | 95.96±2.54 | 96.02±2.61 | 87.43±2.52 | 4.01 | 72.01±41.78 | 72.18±41.74 | 62.84±34.30 | 18.51 | 99.76±0.05 | 99.80±0.05 | 98.89±0.12 | 0.34 |
| SCAR | ✓ | 93.73±0.30 | 93.65±0.09 | 85.18±0.30 | 4.72 | 87.93±0.41 | 87.66±0.04 | 71.36±0.27 | 9.28 | 99.47±0.04 | 99.38±0.02 | 98.40±0.05 | 0.39 |
| **RUAGO** | ✓ | 98.70±0.08 | 91.42±0.58 | 89.89±0.10 | **0.75** | 97.29±0.41 | 82.54±1.11 | 82.48±0.30 | **1.71** | 99.75±0.04 | 99.08±0.04 | 98.81±0.05 | **0.08** |

(b) Results on VGGFace2

| | $\mathcal{D}_r$ free | VGG16 | | | | ResNet18 | | | | ViT | | | |
|---|---|---|---|---|---|---|---|---|---|---|---|---|---|
| | | RA | UA | TA | AVG | RA | UA | TA | AVG | RA | UA | TA | AVG |
| Original | - | 98.13±0.15 | 98.59±0.20 | 95.96±0.10 | - | 99.88±0.02 | 99.88±0.02 | 97.52±0.07 | - | 99.24±0.06 | 99.25±0.15 | 96.79±0.18 | - |
| Retrain | ✗ | 99.91±0.01 | 94.07±0.07 | 94.59±0.27 | 0 | 92.09±0.13 | 85.85±0.11 | 86.22±0.26 | 0 | 98.81±0.10 | 94.03±0.18 | 94.19±0.20 | 0 |
| Bad-T | ✗ | 99.82±0.01 | 96.84±0.82 | 94.12±0.72 | 1.11 | 98.03±0.21 | 79.78±3.95 | 88.92±0.86 | **4.90** | 98.50±0.11 | 93.77±0.31 | 93.72±0.15 | **0.34** |
| SCRUB | ✗ | 100.00±0.00 | 96.69±5.19 | 97.44±0.02 | 1.85 | 97.27±0.15 | 97.26±0.09 | 94.18±0.07 | 8.21 | 99.52±0.04 | 99.30±0.05 | 96.99±0.05 | 2.93 |
| SalUn | ✗ | 99.96±0.02 | 92.60±1.26 | 93.01±0.43 | **1.03** | 98.74±0.16 | 92.09±0.77 | 91.66±0.60 | 6.11 | 98.01±0.33 | 94.83±0.42 | 93.44±0.60 | 0.78 |
| BS | ✓ | 76.58±0.67 | 76.64±0.70 | 78.41±11.33 | 18.98 | 69.50±1.17 | 69.19±1.53 | 65.65±1.30 | 19.94 | 98.63±0.12 | 98.61±0.08 | 95.95±0.04 | 2.17 |
| SSD | ✓ | 97.58±0.38 | 97.59±0.48 | 93.88±0.63 | 2.19 | 83.77±2.68 | 83.88±2.93 | 81.12±2.71 | 5.13 | 97.39±0.58 | 97.39±0.42 | 94.65±0.52 | 1.75 |
| SCAR | ✓ | 97.89±0.41 | 97.93±0.34 | 94.08±0.59 | 14.15 | 88.25±0.88 | 86.99±2.53 | 84.70±0.82 | 2.17 | 97.27±0.08 | 97.20±0.02 | 94.54±0.13 | 1.68 |
| **RUAGO** | ✓ | 98.11±0.04 | 93.61±0.15 | 94.32±0.09 | **0.84** | 90.03±0.26 | 87.56±0.25 | 87.28±0.51 | **1.61** | 97.58±0.11 | 93.32±0.35 | 94.44±0.17 | **0.73** |

# F   Results with 50% deletion

We designate 50% of the entire training dataset as forget set, thereby removing half of the training data. We conduct these experiments on the CIFAR-10 and VGGFace2 datasets. The results are summarized in Table 6. RUAGO demonstrates consistently robust unlearning performance even in large-scale unlearning. On the CIFAR-10 dataset, results are similar to those presented in Section 5.3, where retain-free methods fail to achieve effective unlearning, leading instead to a significant decline in overall model performance. In particular, the Boundary Shrink method demonstrates a substandard outcome with an AVG value of 18.52. Retain-based methods also exhibit suboptimal unlearning performance; for instance, the SCRUB method records an AVG value as high as 3.94 in certain cases. In contrast, RUAGO achieves an AVG value of 0.75 and 0.08 on ResNet18 and ViT models, respectively, outperforming both retain-free and retain-based methods. Similarly, on the VGGFace2 dataset, various baseline methods fail to achieve complete unlearning. Retain-free methods consistently fail to achieve effective unlearning, and even retain-based methods, such as the Bad-T method, exhibit excessive misclassification of $\mathcal{D}_f$ when applied to the ResNet18 model. Conversely, RUAGO not only outperforms all retain-free methods but also achieves lower AVG values than retain-based methods on ResNet18 and VGG16 models. These results highlight RUAGO as a practical and scalable solution for unlearning, effectively removes forget data while consistently maintaining model utility.

# G   Ablation Studies

Table 8 presents the ablation study results on the CIFAR-10, CIFAR-100, TinyImageNet, and VGGFace2 datasets. To assess the utility of $\mathbf{p}_{\mathrm{adv}}$, we perform experiments using hard labels, which lead to UA values significantly lower than TA, indicating over-unlearning. This indicates that most samples in $\mathcal{D}_f$ are misclassified as an outcome counter to the objectives of instance-wise unlearning. If $\mathcal{D}_f$, which should behave as unseen data due to model generalization, is entirely misclassified, it risks triggering the Streisand effect.

Furthermore, unlearning experiments with a generator trained on the TinyImageNet dataset demonstrate effective unlearning performance, highlighting the robustness of RUAGO when employing a generator trained on different OOD datasets. Specifically, we exclude experiments with classification models trained on the TinyImageNet dataset, as the training datasets for both the classification model and generator are identical. The unlearning performance is significantly diminished when a generator is trained from scratch through model inversion rather than pre-trained on an OOD dataset. Interestingly, omitting model inversion occasionally results in slightly better AVG values; however, the overall benefit of incorporating model inversion is evident across different datasets. Lastly, the

Table 7: Performance comparison for unlearning 10% of $\mathcal{D}$ on Stanford Cars dataset.

| | $\mathcal{D}_r$ free | VGG16 | | | | ResNet18 | | | | ViT | | | |
|---|---|---|---|---|---|---|---|---|---|---|---|---|---|
| | | RA | UA | TA | AVG | RA | UA | TA | AVG | RA | UA | TA | AVG |
| Original | - | 99.72±0.04 | 99.63±0.15 | 52.97±1.62 | - | 99.61±0.05 | 99.51±0.15 | 74.85±1.04 | - | 92.67±0.30 | 92.38±0.60 | 77.92±0.49 | - |
| Retrain | ✗ | 99.72±0.04 | 45.38±1.45 | 44.14±1.06 | 0 | 99.51±0.07 | 72.09±1.66 | 71.61±1.06 | 0 | 91.93±0.44 | 75.80±0.85 | 75.96±0.29 | 0 |
| Bad-T | ✗ | 99.60±0.02 | 32.51±11.54 | 49.73±0.31 | 6.19 | 99.42±0.05 | 67.69±6.53 | 72.72±0.62 | **1.87** | 92.31±0.10 | 80.17±1.48 | 77.05±0.16 | **1.95** |
| SCRUB | ✗ | 98.46±0.60 | 48.60±2.93 | 51.17±1.77 | 3.84 | 99.74±0.05 | 75.58±1.44 | 73.63±0.27 | 1.91 | 78.88±40.55 | 69.68±36.38 | 63.81±32.76 | 8.85 |
| SALUN | ✗ | 99.85±0.02 | 47.81±2.43 | 45.62±0.67 | 1.34 | 99.82±0.03 | 44.45±14.09 | 67.64±0.43 | 10.64 | 98.30±0.21 | 73.56±2.18 | 77.12±0.50 | 3.26 |
| BS | ✓ | 69.70±1.47 | 56.98±1.93 | 28.16±0.74 | 19.2 | 99.42±0.04 | 99.39±0.34 | 72.41±0.10 | 9.39 | 89.54±0.67 | 88.97±1.56 | 74.79±0.79 | 5.57 |
| SSD | ✓ | 95.87±3.85 | 95.13±4.53 | 47.49±4.23 | 18.98 | 98.56±8.76 | 97.86±1.16 | 72.60±1.16 | 9.24 | 85.28±5.84 | 80.25±1.33 | 71.07±8.58 | 5.32 |
| SCAR | ✓ | 60.76±2.29 | 57.74±3.22 | 27.71±1.10 | 16.25 | 22.15±9.13 | 21.57±8.28 | 12.20±5.64 | 62.43 | 11.80±7.55 | 12.14±8.02 | 9.56±6.12 | 70.07 |
| **RUAGO** | ✓ | 89.89±0.17 | 45.28±1.29 | 37.51±0.09 | **5.52** | 97.77±0.08 | 69.29±1.14 | 65.97±0.15 | **3.39** | 91.19±0.25 | 76.76±0.65 | 75.70±0.16 | **0.65** |

Table 8: Results of the ablations studies on four datasets.

| | | CIFAR-10 | | | | | VGGFace2 | | | | |
|---|---|---|---|---|---|---|---|---|---|---|---|
| | | RA | UA | TA | MIA | AVG | RA | UA | TA | MIA | AVG |
| VGG16 | **RUAGO** | 99.94±0.02 | 93.38±0.39 | 92.32±0.29 | 0.85±0.00 | 0.282 | 97.44±0.06 | 95.44±0.26 | 94.88±0.17 | 0.69±0.01 | 0.567 |
| | Hard Labels | 96.61±1.23 | 93.93±1.78 | 87.62±1.29 | 0.51±0.04 | 3.146 | 82.98±0.91 | 55.96±0.62 | 79.73±0.65 | 0.59±0.01 | 22.951 |
| | Diff. OOD | 99.93±0.03 | 93.55±0.49 | 92.20±0.17 | 0.85±0.01 | 0.384 | 97.38±0.09 | 95.45±0.28 | 94.82±0.12 | 0.68±0.01 | 0.612 |
| | w/ Init $G_\psi$ | 72.55±16.08 | 72.45±15.63 | 68.22±13.85 | 0.60±0.09 | 24.338 | 1.57±0.21 | 1.60±0.22 | 1.47±0.18 | 0.73±0.02 | 94.287 |
| | w/o MI | 99.95±0.02 | 93.48±0.52 | 92.38±0.16 | 0.85±0.00 | 0.297 | 97.44±0.11 | 95.49±0.28 | 94.91±0.08 | 0.70±0.01 | 0.573 |
| | w/o SD | 99.47±0.11 | 88.93±0.22 | 91.44±0.23 | 0.86±0.00 | 2.133 | 96.75±0.12 | 94.68±0.38 | 94.49±0.19 | 0.71±0.01 | 0.673 |
| ResNet18 | **RUAGO** | 99.34±0.15 | 84.02±1.00 | 84.36±0.26 | 0.48±0.00 | 1.654 | 99.32±0.05 | 96.64±0.22 | 96.41±0.15 | 0.72±0.01 | 0.453 |
| | Hard Labels | 98.12±0.74 | 87.55±2.84 | 82.65±0.99 | 0.37±0.03 | 2.003 | 60.96±1.43 | 31.52±0.57 | 57.12±1.23 | 0.61±0.01 | 48.041 |
| | Diff. OOD | 99.17±0.15 | 84.13±1.40 | 83.98±0.28 | 0.48±0.01 | 1.799 | 99.34±0.05 | 96.61±0.16 | 96.44±0.15 | 0.72±0.01 | 0.442 |
| | w/ Init $G_\psi$ | 27.55±6.74 | 27.15±6.68 | 26.10±6.46 | 0.34±0.22 | 63.961 | 66.27±18.51 | 66.31±18.41 | 64.71±17.47 | 0.46±0.02 | 32.141 |
| | w/o MI | 99.34±0.14 | 84.60±1.06 | 84.38±0.16 | 0.48±0.01 | 1.454 | 99.31±0.06 | 96.64±0.22 | 96.35±0.16 | 0.72±0.01 | 0.476 |
| | w/o SD | 84.08±0.51 | 48.31±0.75 | 68.54±0.38 | 0.73±0.01 | 23.917 | 98.95±0.06 | 95.82±0.26 | 95.97±0.18 | 0.69±0.01 | 0.992 |
| ViT | **RUAGO** | 99.81±0.01 | 98.95±0.16 | 98.95±0.05 | 0.89±0.01 | 0.047 | 98.91±0.02 | 96.94±0.20 | 96.28±0.09 | 0.76±0.01 | 0.425 |
| | Hard Labels | 95.29±0.75 | 93.39±0.92 | 94.18±0.75 | 0.71±0.01 | 4.985 | 88.78±1.53 | 86.23±1.91 | 85.69±1.58 | 0.54±0.01 | 10.431 |
| | Diff. OOD | 99.80±0.01 | 98.86±0.22 | 98.93±0.02 | 0.89±0.00 | 0.075 | 98.91±0.02 | 96.94±0.18 | 96.28±0.10 | 0.76±0.01 | 0.425 |
| | w/ Init $G_\psi$ | 99.54±0.05 | 98.44±0.24 | 98.42±0.06 | 0.87±0.01 | 0.472 | 98.83±0.03 | 96.82±0.17 | 96.10±0.13 | 0.76±0.01 | 0.471 |
| | w/o MI | 99.81±0.01 | 98.94±0.16 | 98.95±0.04 | 0.89±0.00 | 0.052 | 98.91±0.02 | 96.94±0.19 | 96.28±0.13 | 0.76±0.01 | 0.427 |
| | w/o SD | 99.66±0.04 | 98.50±0.26 | 98.61±0.04 | 0.88±0.01 | 0.345 | 98.84±0.03 | 96.80±0.19 | 96.11±0.10 | 0.76±0.01 | 0.457 |

| | | CIFAR-100 | | | | | TinyImageNet | | | | |
|---|---|---|---|---|---|---|---|---|---|---|---|
| | | RA | UA | TA | MIA | AVG | RA | UA | TA | MIA | AVG |
| VGG16 | **RUAGO** | 99.03±0.13 | 67.00±0.98 | 68.70±0.35 | 0.49±0.00 | 3.021 | 99.87±0.02 | 55.65±0.62 | 55.02±0.08 | 0.33±0.01 | 1.913 |
| | Hard Labels | 95.75±0.34 | 25.55±1.17 | 63.16±0.49 | 0.45±0.01 | 19.779 | 98.91±0.10 | 7.97±0.69 | 51.50±0.18 | 0.58±0.01 | 19.293 |
| | Diff. OOD | 99.13±0.13 | 69.24±2.16 | 68.63±0.43 | 0.48±0.01 | 2.264 | - | - | - | - | - |
| | w/ Init $G_\psi$ | 16.12±5.43 | 14.57±4.60 | 13.87±4.14 | 0.40±0.04 | 66.411 | 4.98±0.62 | 4.65±0.54 | 4.17±0.31 | 0.40±0.25 | 81.710 |
| | w/o MI | 99.01±0.14 | 67.22±1.04 | 68.68±0.35 | 0.49±0.00 | 2.964 | 99.87±0.03 | 55.93±0.79 | 54.79±0.06 | 0.33±0.01 | 1.894 |
| | w/o SD | 89.37±0.71 | 51.35±0.56 | 57.60±0.37 | 0.69±0.00 | 15.157 | 94.42±0.52 | 37.44±0.26 | 48.59±0.61 | 0.68±0.00 | 11.941 |
| ResNet18 | **RUAGO** | 99.01±0.18 | 58.90±1.48 | 52.42±0.53 | 0.47±0.01 | 3.086 | 98.66±0.15 | 41.52±1.10 | 39.48±0.22 | 0.36±0.01 | 3.101 |
| | Hard Labels | 91.93±0.59 | 14.58±2.14 | 45.01±0.40 | 0.76±0.03 | 22.690 | 85.88±0.92 | 4.50±0.47 | 29.76±0.39 | 0.87±0.02 | 41.103 |
| | Diff. OOD | 99.22±0.21 | 56.25±1.62 | 52.65±0.51 | 0.45±0.02 | 3.822 | - | - | - | - | - |
| | w/ Init $G_\psi$ | 2.40±0.82 | 2.34±0.65 | 2.15±0.69 | 0.42±0.13 | 70.898 | 2.50±0.31 | 2.79±0.36 | 2.20±0.30 | 0.20±0.29 | 62.339 |
| | w/o MI | 99.00±0.14 | 59.65±1.03 | 52.19±0.52 | 0.48±0.01 | 2.918 | 98.59±0.17 | 41.11±1.43 | 39.54±0.31 | 0.36±0.00 | 3.243 |
| | w/o SD | 76.26±0.89 | 48.76±0.20 | 39.34±0.13 | 0.68±0.01 | 18.411 | 43.52±0.97 | 24.08±0.27 | 21.53±0.43 | 0.82±0.01 | 45.952 |
| ViT | **RUAGO** | 97.65±0.06 | 94.64±0.33 | 92.10±0.06 | 0.74±0.00 | 1.228 | 95.98±0.03 | 92.44±0.23 | 90.56±0.06 | 0.77±0.00 | 0.773 |
| | Hard Labels | 90.31±0.47 | 86.38±0.67 | 84.52±0.32 | 0.63±0.01 | 7.302 | 86.11±0.27 | 80.45±0.46 | 81.31±0.19 | 0.62±0.01 | 10.064 |
| | Diff. OOD | 97.59±0.08 | 94.65±0.29 | 92.00±0.15 | 0.74±0.00 | 1.286 | - | - | - | - | - |
| | w/ Init $G_\psi$ | 95.14±0.25 | 92.05±0.58 | 89.32±0.10 | 0.72±0.01 | 2.200 | 95.72±0.06 | 92.38±0.24 | 90.24±0.16 | 0.77±0.00 | 1.712 |
| | w/o MI | 97.64±0.07 | 94.67±0.34 | 92.08±0.12 | 0.75±0.00 | 1.249 | 95.99±0.03 | 92.45±0.26 | 90.59±0.07 | 0.77±0.00 | 0.765 |
| | w/o SD | 95.64±0.15 | 92.32±0.45 | 89.81±0.15 | 0.72±0.01 | 1.890 | 95.76±0.05 | 92.38±0.24 | 90.29±0.12 | 0.77±0.00 | 0.919 |

importance of including sample difficulty is apparent, as omitting this strategy, particularly for the TinyImageNet dataset, leads to complete unlearning failure.

To further demonstrate the effectiveness of our approach, we conduct experiments on a fine-grained dataset where class-level similarities make discrimination particularly challenging. The experimental results on the Stanford Cars dataset [59] are presented in Table 7. The Stanford Cars dataset poses challenges for unlearning due to many similar classes and features. Nevertheless, our method shows consistently strong performance, achieving an AVG of 0.65 on the ViT model, outperforming all baselines.

# H  Class-wise Unlearning Scenario Experiments

As an extension of our main experiments, we explore class-wise unlearning to assess how well unlearning methods can eliminate information tied to specific semantic categories. Unlike the instance-wise unlearning scenario, which aims to delete individual instance samples independently of their classes, class-wise unlearning involves erasing all information about a specific class. Hence, an ideal class-wise unlearning outcome for a classification model should result in entirely incorrect predictions

Table 9: Sensitivity analysis of inversion loss weights $(\gamma_{\mathrm{adv}}, \gamma_{\mathrm{bn}}, \gamma_{\mathrm{cls}})$ on CIFAR-10 with ResNet18.

| $\gamma_{\mathrm{adv}}$ | $\gamma_{\mathrm{bn}}$ | $\gamma_{\mathrm{cls}}$ | RA | UA | TA |
|---|---|---|---|---|---|
| **0** | 1 | 1 | 99.25 | 96.84 | 96.29 |
| **1** | 1 | 1 | 99.26 | 96.92 | 96.27 |
| **2** | 1 | 1 | 99.24 | 96.86 | 96.29 |
| **3** | 1 | 1 | 99.24 | 96.84 | 96.22 |
| 0 | **2** | 1 | 99.25 | 96.92 | 96.25 |
| 0 | **3** | 1 | 99.25 | 96.86 | 96.23 |
| 0 | 1 | **2** | 99.25 | 96.82 | 96.26 |
| 0 | 1 | **3** | 99.24 | 96.82 | 96.23 |

Table 10: Sensitivity analysis of main loss weights $(\gamma_1, \gamma_2)$ on CIFAR-10 with ResNet18.

| $\gamma_1$ | $\gamma_2$ | RA | UA | TA |
|---|---|---|---|---|
| **0** | 0.01 | 95.34 | 95.29 | 91.65 |
| **0.15** | 0.01 | 99.24 | 96.86 | 96.29 |
| **0.5** | 0.01 | 99.28 | 96.45 | 96.39 |
| **1** | 0.01 | 99.27 | 96.35 | 96.34 |
| **1.5** | 0.01 | 99.22 | 96.28 | 96.32 |
| 0.15 | **0** | 98.59 | 96.00 | 95.49 |
| 0.15 | **0.01** | 99.24 | 96.86 | 96.29 |
| 0.15 | **0.5** | 98.54 | 98.91 | 95.30 |
| 0.15 | **1** | 97.92 | 98.22 | 94.49 |
| 0.15 | **1.5** | 97.49 | 97.71 | 93.90 |

for the targeted class. Table 13 presents the performance of various unlearning methods across three models trained on the CIFAR-10 dataset.

Specifically, for TRA and TUA, the reported accuracies correspond to the test retain set and test forget set, respectively, based on the targeted class within the test set $\mathcal{D}_t$. In alignment with the Retrain model, RA and TRA should remain as high as possible, while UA and TUA should be close to 0, indicating complete deletion.

All baseline methods generally exhibit robust class-wise unlearning performance; however, certain limitations are observed. For instance, the Bad Teaching method records high RA and TRA values, demonstrating that model utility remains intact, yet the elevated UA and TUA scores reveal incomplete deletion. SCRUB exhibits highly unstable unlearning results across different seeds, particularly for the VGG16 and ViT model. SalUn outperforms the other two baselines that utilize the retain set, showcasing more stable unlearning performance but still falling short of complete deletion.

Turning to retain-free methodologies, Boundary Shrink not only fails to preserve model utility but also does not accomplish complete deletion. In contrast, SSD and SCAR deliver impressive unlearning performance without relying on a retain set. However, SSD fails to handle larger models, as shown by its poor results on ViT. SCAR similarly struggles with large models and incurs substantial training time costs, as evidenced by the ResNet18 scenario, making it inefficient since it requires significant computation for each unlearning request. In contrast, our approach involves training the generator on out-of-distribution (OOD) data, allowing us to prepare for unlearning requests proactively before they are made.

In our proposed method, RUAGO, we filter out the generated outputs $\tilde{x}$ that are predicted as belonging to the class slated for deletion. While not exhibiting the highest performance metrics, our approach provides adequate and stable unlearning performance. This characteristic stems from the core design of RUAGO: its main signal, the adversarial soft target, is optimized for reshaping decision boundaries locally around individual samples, making it highly effective for instance-wise unlearning. In contrast, class-wise unlearning requires a more global ob-

Table 11: Effect of alignment learning rate (lr). Excessive alignment strength leads to collapse.

| LR | RA | UA | TA |
|---|---|---|---|
| 5.0e-05 | 99.14 | 97.12 | 96.09 |
| 1.0e-04 | 99.24 | 96.84 | 96.29 |
| 5.0e-04 | 98.21 | 96.62 | 95.71 |
| 1.0e-03 | 94.20 | 96.49 | 92.87 |
| 5.0e-03 | 11.63 | 15.47 | 12.14 |
| 1.0e-02 | 2.95 | 3.11 | 3.07 |

jective of erasing an entire semantic concept across the input space. To maintain a consistent and unified pipeline, we applied the same mechanism to both scenarios. This design choice led to effective unlearning, especially on VGG-16 and ViT where UA/TUA values were near-zero, though it resulted in residual information in some specific model-setting combinations. Notably, this highlights a crucial trade-off. Other retain-free methods such as SSD and SCAR, which perform well in class-wise deletion, tend to struggle significantly with instance-wise unlearning, larger models, or high computational costs. In contrast, RUAGO maintains commendable and balanced performance across both class-wise and instance-wise unlearning scenarios. This versatility underscores the applicability of RUAGO to diverse unlearning challenges, demonstrating its robustness and efficacy in various operational contexts.

# I  OOD Datasets and Generator Pretraining

To assess distributional differences between classification datasets and generator training sets, we embed each image with a CLIP [65] encoder and compute the Fréchet Distance (FD):

$$\text{FD}(\mathcal{X}, \mathcal{Y}) = \|\boldsymbol{\mu}_\mathcal{X} - \boldsymbol{\mu}_\mathcal{Y}\|^2 + \text{Tr}\big(\boldsymbol{\Sigma}_\mathcal{X} + \boldsymbol{\Sigma}_\mathcal{Y} - 2\,(\boldsymbol{\Sigma}_\mathcal{X}\boldsymbol{\Sigma}_\mathcal{Y})^{\frac{1}{2}}\big),$$

where $(\boldsymbol{\mu}_\mathcal{X}, \boldsymbol{\Sigma}_\mathcal{X})$ and $(\boldsymbol{\mu}_\mathcal{Y}, \boldsymbol{\Sigma}_\mathcal{Y})$ represent the means and covariances of the features extracted from two datasets, $\mathcal{X}$ and $\mathcal{Y}$, respectively. Lower FD indicates more similar distributions. Figure 5 shows these FD scores as a confusion matrix. COCO diverges markedly from all four classification datasets, and TinyImageNet has an exceptionally high FD with VGGFace2. These observations confirm that COCO and TinyImageNet are valid OOD sources under our easy-access assumption.

Table 12 summarizes ablation results using generators pretrained on three datasets: in-domain CIFAR-10 and two OOD sources (COCO, TinyImageNet). Although a generator pretrained on the in-domain data (CIFAR-10) yields the best unlearning performance, it presents a critical flaw: to prevent information leakage, it must be discarded after a single unlearning task. This single-use nature makes it impractical for scenarios requiring sequential unlearning requests and introduces substantial overhead. This highlights a fundamental challenge in retain-free unlearning: a mechanism is needed to approximate the distribution of data that should be retained. We posit that an OOD-trained generator serves as an efficient and straightforward surrogate for this purpose. The importance of this pretraining step is demonstrated in Table 8, where using an untrained, randomly initialized generator results in a significant performance drop. Therefore, we focus our main approach on OOD-pretrained models.

We further evaluate an extreme OOD case by pretraining on VGGFace2. Despite its much smaller size, making it less readily accessible and poses challenges for generator training, the VGGFace2 generator only slightly underperforms the COCO model. Still, it outperforms all retain-free baselines from Table 1, demonstrating robustness even in challenging worst-case scenarios.

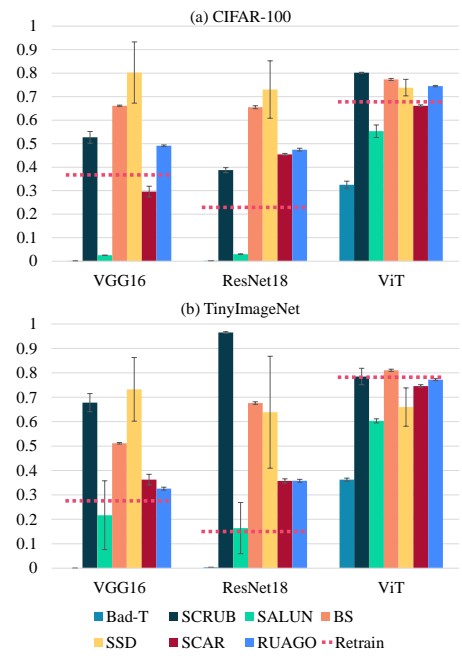

Figure 4: MIA results for each unlearning method. The red dotted line represents the Retrain model; it is best to be closer to this line.

We use a StyleGAN2-ADA [66, 67] backbone for these experiments. While this requires a pretraining step, we consider its cost a practical trade-off for effective unlearning under the strict retain-free constraint. Training on COCO took ≈4 hours and on TinyImageNet ≈3 hours (32×32 resolution on our GPU), with generated samples shown in Figures 6 and 7. To mitigate this dependency in future work, one could explore the use of off-the-shelf generative models, assuming their training data does not introduce other privacy issues. We expect higher-resolution training to boost unlearning performance further.

# J  Limitations and Future Work

We can point out several limitations of RUAGO. Initially, our experiments utilize a generator trained on datasets with rich feature content. Generally, numerous open-source datasets with rich features are easily accessible, making them easy to obtain. However, there may be extreme scenarios where such datasets are not accessible. In such situations, it remains uncertain whether unlearning would be complete when employing a generator trained on simpler datasets, such as those with single-channel images, to unlearn classification models trained on complex, high-feature datasets. Additionally, our experiments are limited to supervised image classification models. However, we believe that the fundamental design principles of RUAGO, particularly curriculum-based training over OOD samples,

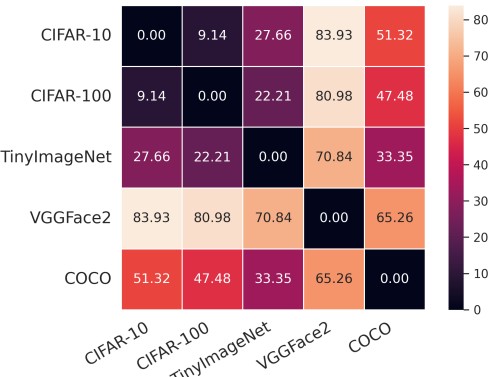

Figure 5: Confusion matrix of FD scores across all datasets.

Table 12: Unlearning performance across different generator training datasets.

| Model | | Generator CIFAR-10 | VGGFace2 | COCO |
|---|---|---|---|---|
| VGG16 | RA | 99.99±0.01 | 99.70±0.09 | 99.94±0.02 |
| | UA | 93.12±0.27 | 91.36±0.50 | 93.38±0.39 |
| | TA | 92.79±0.05 | 91.33±0.22 | 92.32±0.29 |
| | AVG | 0.11 | 1.28 | 0.28 |
| ResNet18 | RA | 99.94±0.01 | 96.60±0.30 | 99.34±0.15 |
| | UA | 84.95±0.61 | 87.28±0.63 | 84.02±1.00 |
| | TA | 86.20±0.24 | 81.57±0.37 | 84.36±0.26 |
| | AVG | 0.69 | 2.78 | 1.65 |
| ViT | RA | 99.83±0.02 | 98.89±1.47 | 99.81±0.01 |
| | UA | 98.98±0.11 | 99.05±0.09 | 98.95±0.16 |
| | TA | 99.00±0.01 | 98.87±0.05 | 98.95±0.05 |
| | AVG | 0.05 | 0.35 | 0.05 |

adversarial soft-target regularization, and generator-teacher alignment, are not inherently tied to this specific domain. For instance, in natural language processing (NLP), these principles could be adapted by replacing the image generator with a small language model, defining sample difficulty via token-level cross-entropy, and performing alignment in the hidden feature space rather than the pixel space. Adversarial soft-targets could similarly be constructed through perturbations applied to token logits. Nonetheless, we do not claim that our method can be directly applied to generative models. Adapting RUAGO to architectures like GANs, diffusion models, or large language models would require careful redesign and new, task-specific metrics. In summary, this work establishes our method in a foundational supervised setting to analyze the core mechanics of unlearning, while acknowledging that extending it to other domains is feasible but requires further investigation.

Consequently, future work should explore more versatile methodologies, including generators trained on low-quality images or even randomly initialized generators that do not rely on OOD data. On the theoretical front, our current analysis offers a generalization bound for the curriculum learning aspect, which helps explain how scheduling sample difficulty stabilizes the unlearning process. However, a unified theoretical framework that encompasses all components of RUAGO, including the adversarial soft-target mechanism, has not yet been established and remains an important direction for future work. Another key direction for future research is to realize the roadmap for adapting RUAGO to other domains. This includes empirically verifying its effectiveness in tasks such as object detection and NLP, guided by the principles outlined above. Expanding the applicability of RUAGO is an important avenue for future research.

Table 13: Class-wise unlearning performance of baseline methods and our proposed RUAGO on the CIFAR-10 dataset.

| | | RA | UA | TRA | TUA | AVG | MIA | RTE |
|---|---|---|---|---|---|---|---|---|
| | | VGG16 | | | | | | |
| Retrain | ✗ | 100.00 ± 0.00 | 0.00 ± 0.00 | 92.74 ± 0.17 | 0.00 ± 0.00 | 0 | 0.41 ± 0.02 | 869 |
| Bad-T | ✗ | 100.00 ± 0.00 | 18.12 ± 35.01 | 92.92 ± 0.13 | 15.74 ± 30.52 | 8.51 | 0.00 ± 0.00 | 65 |
| SCRUB | ✗ | 80.56 ± 38.69 | 0.00 ± 0.00 | 73.28 ± 34.66 | 0.00 ± 0.00 | 9.72 | 0.26 ± 0.15 | 112 |
| SalUn | ✗ | 100.00 ± 0.00 | 0.81 ± 1.23 | 92.42 ± 0.21 | 0.70 ± 0.92 | 0.46 | 0.00 ± 0.00 | 65 |
| BS | ✓ | 94.17 ± 0.35 | 29.98 ± 0.91 | 85.26 ± 0.40 | 28.32 ± 0.50 | 17.9 | 0.19 ± 0.01 | 62 |
| SSD | ✓ | 100.00 ± 0.00 | 0.00 ± 0.00 | 93.09 ± 0.02 | 0.00 ± 0.00 | 0.09 | 0.00 ± 0.00 | 15 |
| SCAR | ✓ | 99.66 ± 0.05 | 0.93 ± 0.03 | 91.46 ± 0.14 | 0.60 ± 0.25 | 0.79 | 0.00 ± 0.00 | 143 |
| **RUAGO** | ✓ | 97.77 ± 0.21 | 0.02 ± 0.04 | 90.18 ± 0.23 | 0.04 ± 0.09 | 1.21 | 0.07 ± 0.02 | 119 |
| | | ResNet18 | | | | | | |
| Retrain | ✗ | 99.99 ± 0.01 | 0.00 ± 0.00 | 86.56 ± 0.24 | 0.00 ± 0.00 | 0 | 0.34 ± 0.02 | 1,917 |
| Bad-T | ✗ | 99.98 ± 0.00 | 19.98 ± 44.36 | 86.62 ± 0.10 | 16.76 ± 37.48 | 9.2 | 0.00 ± 0.00 | 81 |
| SCRUB | ✗ | 100.00 ± 0.00 | 0.00 ± 0.00 | 86.76 ± 0.15 | 0.00 ± 0.00 | 0.05 | 0.01 ± 0.00 | 67 |
| SalUn | ✗ | 100.00 ± 0.00 | 7.39 ± 8.13 | 86.49 ± 0.13 | 5.76 ± 7.96 | 3.31 | 0.00 ± 0.00 | 67 |
| BS | ✓ | 85.79 ± 0.66 | 19.40 ± 0.49 | 75.02 ± 0.46 | 18.62 ± 0.54 | 15.94 | 0.32 ± 0.00 | 82 |
| SSD | ✓ | 99.99 ± 0.00 | 0.00 ± 0.00 | 86.70 ± 0.04 | 0.00 ± 0.00 | 0.04 | 0.00 ± 0.00 | 15 |
| SCAR | ✓ | 98.89 ± 0.25 | 0.95 ± 0.03 | 84.38 ± 0.22 | 0.42 ± 0.23 | 1.16 | 0.12 ± 0.01 | 3,793 |
| **RUAGO** | ✓ | 94.95 ± 0.37 | 5.79 ± 0.51 | 81.24 ± 0.32 | 6.00 ± 0.79 | 5.54 | 0.19 ± 0.01 | 489 |
| | | ViT | | | | | | |
| Retrain | ✗ | 99.85 ± 0.03 | 0.00 ± 0.00 | 98.90 ± 0.07 | 0.00 ± 0.00 | 0 | 0.04 ± 0.01 | 4,856 |
| Bad-T | ✗ | 99.78 ± 0.03 | 12.45 ± 4.27 | 98.95 ± 0.08 | 11.44 ± 5.39 | 6 | 0.00 ± 0.00 | 4,592 |
| SCRUB | ✗ | 99.75 ± 0.18 | 80.33 ± 43.53 | 98.88 ± 0.22 | 80.34 ± 43.18 | 40.20 | 0.74 ± 0.24 | 3,656 |
| SalUn | ✗ | 99.99 ± 0.00 | 2.33 ± 0.51 | 98.97 ± 0.06 | 2.12 ± 0.60 | 1.16 | 0.00 ± 0.00 | 2,832 |
| BS | ✓ | 94.45 ± 0.81 | 55.16 ± 7.65 | 93.45 ± 0.77 | 54.54 ± 7.76 | 30.14 | 0.05 ± 0.03 | 522 |
| SSD | ✓ | 84.77 ± 0.29 | 1.98 ± 0.75 | 83.70 ± 0.33 | 1.88 ± 0.69 | 8.54 | 0.06 ± 0.00 | 593 |
| SCAR | ✓ | 23.55 ± 9.78 | 0.00 ± 0.00 | 23.35 ± 9.48 | 0.00 ± 0.00 | 37.96 | 0.27 ± 0.22 | 265 |
| **RUAGO** | ✓ | 99.60 ± 0.05 | 0.78 ± 0.13 | 98.59 ± 0.10 | 0.76 ± 0.09 | 0.53 | 0.02 ± 0.01 | 524 |

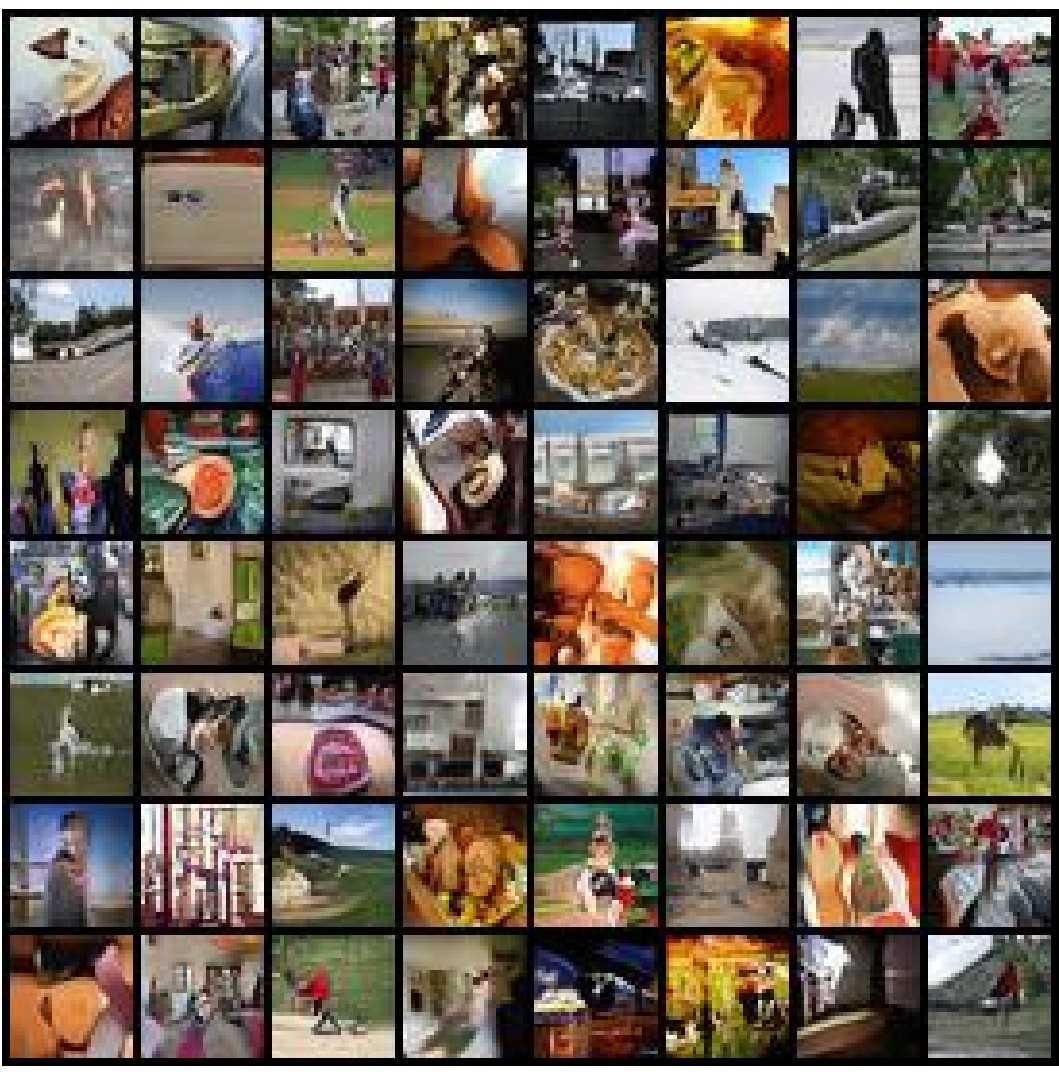

Figure 6: Sample images generated by the generator trained on the COCO dataset.

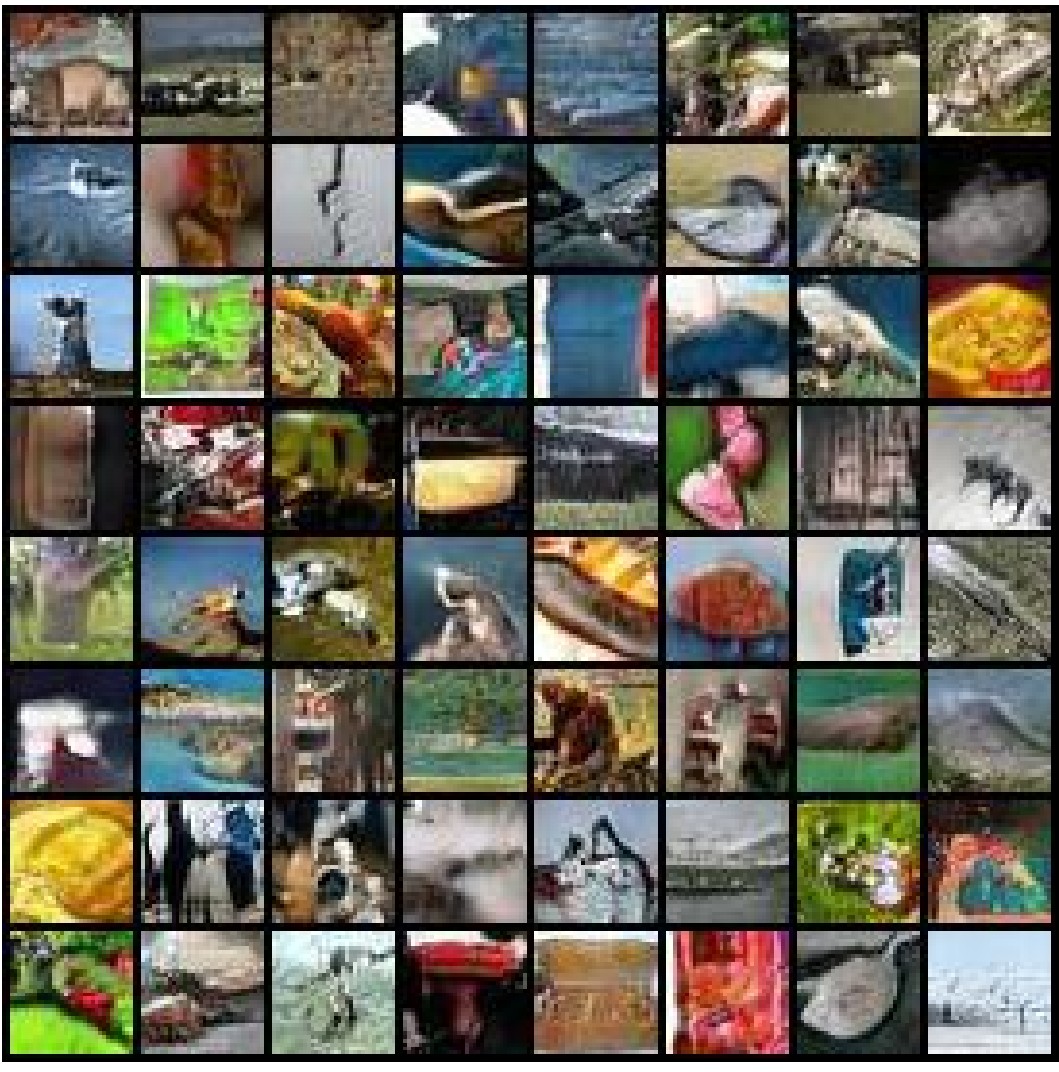

Figure 7: Sample images generated by the generator trained on the TinyImageNet dataset.

