# OpenReview forum: "RUAGO: Effective and Practical Retain-Free Unlearning via Adversarial Attack and OOD Generator"
_NeurIPS.cc/2025/Conference — NeurIPS 2025 poster_

### Official Review · Reviewer_CKem · 2025-06-19

**Clarity:** 3
**Significance:** 3
**Originality:** 4
**Rating:** 5
**Confidence:** 3

**Summary:**

The paper introduces **RUAGO**, a retain-free unlearning method that uses soft-label adversarial examples to achieve unlearning while avoiding over-unlearning. To preserve model performance, the paper leverages synthetic data from an OOD-trained generator, which is aligned with the original data distribution through inversion-based alignment.

**Questions:**

I would like to point out two trivial questions (**TQ**), one minor question (**mQ**), and one major question (**MQ**) that, if addressed, would lead me to reconsider the scores for the quality and clarity of this work.

**TQ1:** In Theorem 1, $R(\cdot)$ is not defined; it becomes clear that it refers to the true risk function only in the proof. Readability would be improved if terms were defined as soon as they are introduced. For the same reason, the authors should define $E$ and $E_{g}$ as they appear in Algorithm 1.

**TQ2:** In the proof of Theorem 1, on a first read, it is not immediately clear that $\epsilon_t > 0$. The authors should emphasize this fact to improve readability.

**mQ1:** The proof of Theorem 1 contains one unclear step that I would like the authors to clarify. In lines 997–998, the authors define $\epsilon_0$, which is then used in lines 999-1000. However, it's unclear why the authors use $\leq$ instead of $=$. While this is not a flaw per se - since the majority argument in lines 1001-1002 would lead to the same conclusion - it is not clear why the inequality is used rather than an equality.

**MQ1:** My main concern regarding the proposed method is its efficacy in _truly_ forgetting information. Specifically, I suspect that the realignment process, where the updated model is aligned with the original data distribution by matching the generator's distribution to that of the original model, might inadvertently reintroduce information about the forget set. This concern is supported by several observations: first, the method often outperforms the retain model in UA performance, and sometimes even in TA performance; second, the MIA score is occasionally higher than that of the retain model; third, the method's performance appears sensitive to the choice of synthetic data generator. It would strengthen the work significantly if the authors could discuss this potential weakness in more detail, as incomplete unlearning undermines both the validity of the method and the overall contribution of the paper. Specifically, when $\gamma_1$ and $\gamma_2$ are kept frozen, does a strong alignment of the generator to the original data distribution hinder the unlearning process?

**Ethical Concerns:**

["NO or VERY MINOR ethics concerns only"]

**Final Justification:**

I was already positive about this submission; plus, the authors have satisfactorily addressed all of my concerns in the rebuttal. In my view, this is a valuable contribution. That said, I remain open to discussing my assessment with the AC and fellow reviewers.

**Limitations:**

Yes. The authors have addressed some of the limitations of their work in the supplementary material (see Section J - "Limitations and Future Work").

**Paper Formatting Concerns:**

Nothing to report.

**Quality:**

2

**Strengths And Weaknesses:**

**Strengths**

*S1 (Soundness).* All claims are well motivated and supported;

*S2 (Novelty).* Original method for performing machine unlearning;

*S3 (Performance).* RUAGO exhibits improved test accuracy compared to other unlearning methods;

*S4 (Clarity).* The paper is clearly written.

**Weaknesses**

*W1.* The work does not provide guarantees regarding the success of the unlearning process. This is the main weakness of the proposed method, as the effectiveness of unlearning depends on the quality of the synthetic data generation. However,  this is a limitation shared by other state-of-the-art methods.

---

> ### Author Rebuttal · Authors · 2025-07-31
>
> **Thank you for Reviewer CKem's insightful comments on our work.**
>
> **[W1] Dependency on synthetic data quality.**
>
> Thank you for your thoughtful comment. We fully agree that performance can depend on the generator. In RUAGO, we try to minimize the generator dependency via adversarial soft targets (grounded on the forget set), an easy-to-hard schedule (down-weighting noisy synthetic samples early), and inversion-based alignment with filtering. To show robustness, we trained a low-fidelity $32\times 32$ GAN on OOD data, which is far simpler than the latest generative models. Even under this constrained setup, RUAGO still outperformed prior retain-free baselines and approached retain-based utility, demonstrating the feasibility of our framework. In the final version, we will incorporate the reviewer’s comments by conducting a brief sensitivity analysis regarding the generator’s complexity and data quality. However, we believe and demonstrate that the current setting is sufficient to achieve good performance with the simple generator.
>
> **[TQ1&2] Notation clarification**
>
> We thank the reviewer for requesting clarification on the notation. We will fully incorporate and clarify the questions.
> More specifically, we define the true risk by $R(\mathcal{F})=\mathbb{E}\_{(x,y)\sim\mathcal{D}}\bigl[\ell\bigl(\mathcal{F}(x),y\bigr)\bigr],$ and the empirical risk by $\hat{R}\_n(\mathcal{F})=\frac1n\sum_{i=1}^{n}\ell\bigl(\mathcal{F}(x\_i),y\_i\bigr).$ In Algorithm 1, $E$ denotes the unlearning epochs and $E_g$ denotes the generator training epochs, respectively.
> We will also make the positivity of $\epsilon_t$ more explicitly as follows: We define $\epsilon_t = C\,\sqrt{\frac{d_t\bigl(\log(n/d_t)+\log(1/\delta)\bigr)}{n}}$, where C>0 is a universal constant from the VC uniform‑convergence bound, $d_t \ge 1$ is the (finite) VC‑dimension of the stage‑t hypothesis space, $n\ge 1$ is the sample size, and $\delta\in(0,1)$, respectively. Hence $\epsilon_t>0$. In the final version, we will clarify all the relevant definitions. Thanks again for your thoughtful comments!
>
> **[mQ1] Clarification in Theorem 1’s proof**
>
> Thank you for the comments and pointing out typos. We are sorry that there are notational errors on our side. We will fully fix and clarify the notations as follows: Since we define $\epsilon\_0 = R(\mathcal{F}\_1)-R(\mathcal{F}^\*)$, the line 999–1000 derivation should use an equality rather than an inequality:
> $$
> R(\mathcal{F}\_T)-R(\mathcal{F}^\*) \;=\; \bigl[\,R(\mathcal{F}\_T)-R(\mathcal{F}\_1)\,\bigr] + \bigl[\,R(\mathcal{F}\_1)-R(\mathcal{F}^\*)\,\bigr] \;=\; \bigl[\,R(\mathcal{F}\_T)-R(\mathcal{F}\_1)\,\bigr] + \epsilon_0.
> $$
> We will correct "$\le$" to "$=$" in the final version and add a note ("equality holds by the definition of $\epsilon\_0$") to clarify.
>
> For lines 1001–1002, the inequality is intended and follows from an add‑and‑subtract decomposition combined with the VC uniform‑convergence bound:
> $$
> R(\mathcal{F}\_t)-R(\mathcal{F}\_{t-1})
> = \bigl[R(\mathcal{F}\_t)-\hat R_n(\mathcal{F}\_t)\bigr]
>  +\bigl[\hat R\_n(\mathcal{F}\_t)-\hat R\_n(\mathcal{F}\_{t-1})\bigr]
>  +\bigl[\hat R\_n(\mathcal{F}\_{t-1})-R(\mathcal{F}\_{t-1})\bigr]
> \le \epsilon\_t + 0 + \epsilon\_{t-1}
> = \epsilon\_t+\epsilon\_{t-1}.
> $$
> The first and third terms are each bounded by $\epsilon\_t$ and $\epsilon\_{t-1}$, respectively, via the standard VC uniform-convergence bound. Combining these bounds yields an upper bound of $\epsilon\_t + \epsilon\_{t-1}$ for their sum. The middle term, $\hat{R}\_n(\mathcal{F}\_t) - \hat{R}\_n(\mathcal{F}\_{t-1})$, is non-positive because the curriculum uses nested classes $\mathcal{H}\_{t-1} \subseteq \mathcal{H}\_t$ and $\mathcal{F}\_t$ minimizes $\hat{R}\_n$ over $\mathcal{H}\_t$; hence we safely upper-bound it by $0$ (written “$+\,0$” for brevity). The inequality is stated for $t\ge 2$, so $\epsilon\_0$ does not enter this step.
> To clarify the notation, we will rename $\epsilon\_0$ to $\epsilon$ and fix the remaining minor notation accordingly in the final version.
>
>
> [MQ1] **Realignment and Re‑injection Risk**
>
> We appreciate the reviewer's concern that alignment could inadvertently re‑inject information from the forget set. We treat this as a primary risk and design both the objective and the pipeline to suppress this issue. The goal of the alignment stage is not to reconstruct specific forgotten samples, but to make synthetic images interpretable in the teacher's representation space so that distillation can be stable. Concretely, we (i) push predictions on the forget set away via the forget loss $\mathcal{L}\_f$, and (ii) in class‑wise unlearning, filter out generator outputs that are classified as the to‑be‑deleted class, preventing re‑injection of that concept.
>
> To probe the effect of alignment strength, we fix $\gamma\_1,\gamma\_2$ and vary a proxy for alignment strength for the learning rate as the alignment component (ResNet‑18/VGGFace2). As shown in the table below, we observe that moderate alignment (e.g., $\mathrm{lr}=10^{-4}$) keeps RA/UA/TA stable, showing that the teacher's representation is emulated without re‑introducing the forget information. In contrast, overly strong alignment (e.g., $\mathrm{lr}\ge 10^{-3}$) induces a collapse in all metrics (UA as well as RA/TA), being consistent with optimization instability rather than selective re‑injection.
>
> | lr       | RA      | UA      | TA      |
> |----------|---------|---------|---------|
> | 5e‑05  | 99.1369 | 97.1246 | 96.0906 |
> | 1e‑04  | 99.2415 | 96.8387 | 96.2874 |
> | 5e‑04  | 98.2121 | 96.6201 | 95.7109 |
> | 1e‑03  | 94.1991 | 96.4856 | 92.8702 |
> | 5e‑03  | 11.6317 | 15.4700 | 12.1361 |
> | 1e‑02  | 2.9518  | 3.1108  | 3.0657  |
>
> Regarding the reviewer's observations: (a) UA (forget‑set accuracy) being slightly higher than retrain in a few cases is not ideal, but the gap is small ($ \sim $ 1% on multi‑seed averages) and differing trends are visible in other methods under the same protocol (e.g., SCRUB, SalUn; see Tables 1 and 4). Possible explanations for this include a small generalization gain inherited from the original model that trained with the forget set, mild underfitting in the retrain model alleviated by additional training, or modest diversity injected by synthetic data. (b) For MIA, we note that in challenging cases (e.g., CIFAR‑100 with smaller models such as ResNet‑18), occasional increases can appear. Overall, RUAGO's gap to retrain is comparable to or smaller than strong baselines (Figures 3–4), suggesting no systematic privacy regression. (c) On generator sensitivity, our experiments intentionally used a lightweight low-resolution $32\times 32$ GAN trained on OOD data. Yet, RUAGO outperformed prior better retain‑free methods that used higher‑resolution, more diverse generators, demonstrating much less generated dataset dependency of our approach.
>
> In summary, with $\gamma_1,\gamma_2$ fixed, tuning alignment within a safe range yields no quantitative signs of re‑injection, whereas excessive alignment causes global collapse, not targeted "re‑learning." Across key metrics, RUAGO remains close to retrain and consistently closer than other retain‑free baselines. Thus, distribution alignment in RUAGO does not materially induce re‑injection; rather, $\mathcal{L}_f$ and the output filtering mechanism mitigate potential leakage, enabling a stable deletion–utility balance. We will make these points explicit in the final version.

---

> > ### Comment · Reviewer_CKem · 2025-08-01
> > **Response to Authors' Rebuttal**
> >
> > I would like to thank the authors for addressing my questions, particularly those related to the clarification of **TQ1**, **TQ2**, and **mQ1**.
> >
> > However, regarding **MQ1**, I still find the setup somewhat unclear and would like to follow up with a few points.
> >
> > First, when referring to the learning rate as a proxy for alignment strength, are you specifically referencing the learning rate used during the generator's alignment phase? If so, I would caution that a higher learning rate does not necessarily imply stronger alignment; in fact, it may lead the generator to diverge or settle in poorly aligned regions.
> >
> > Assuming this interpretation is correct, I would suggest re-running the experiment by fixing the learning rate (e.g., at $5e$-$5$, $1e$-$4$, or $5e$-$4$) and reporting how RA, UA, and TA evolve over the course of the alignment epochs. Specifically, a table with columns such as:
> >
> > Epoch | RA | UA | TA
> >
> > would be informative.
> >
> > Even if limited to a single experimental setting, I would appreciate seeing such a table within the rebuttal window.
> >
> > Second, if the generator is indeed poorly aligned, one would expect a noticeable drop in performance. Yet, the small difference between the first two rows of the table seems to suggest that alignment might not be contributing meaningfully.
> >
> > The epoch-wise evolution table would help clarify both of these issues.

---

> > > ### Author Response · Authors · 2025-08-06
> > >
> > > We thank the reviewer for the helpful follow-up.  We fixed the generator-alignment LR and varied only the number of alignment epochs $E_g$, reporting RA/UA/TA on VGGFace2 with VGG16, ResNet18, and ViT (table below). Across all three models, the metrics remain stable as $E_g$ increases. This indicates that alignment operates in a safe regime, stabilizing distillation without re-injecting the forget information.  For VGG16 and ResNet18, we observe small but consistent gains in RA and TA as $E_g$ grows, suggesting that extra alignment can improve utility without harming deletion. In practice, we use a small number of epochs for runtime efficiency, since gains beyond a few epochs are marginal. The forget loss $L_f$ operates on the unlearning model and serves as a safeguard. It penalizes any tendency of the model to fit the forget set and pushes predictions away from it, preventing such drift from propagating to the final model.
> > >
> > > Regarding the small difference between the first two rows in the earlier LR table, both settings lie in a conservative/stable regime, hence the similar final metrics. The epoch-wise results here show that extending alignment does not systematically worsen UA or utility. Moreover, in an ablation without alignment (Sec. 5.4), utility and/or deletion metrics decline. Even lightweight alignment consistently stabilizes distillation and preserves RA/TA without increasing UA. We will include this table and state the practical guideline in the final version.
> > >
> > > | $E_g$ | VGG16 RA (±) | VGG16 UA (±) | VGG16 TA (±) | ResNet18 RA (±) | ResNet18 UA (±) | ResNet18 TA (±) | ViT RA (±) | ViT UA (±) | ViT TA (±) |
> > > |------:|:-------------:|:------------:|:------------:|:---------------:|:---------------:|:---------------:|:----------:|:----------:|:----------:|
> > > | 10     | 97.44 ± 0.06  | 95.44 ± 0.26 | 94.88 ± 0.17 | 99.32 ± 0.05    | 96.64 ± 0.22    | 96.41 ± 0.15    | 98.91 ± 0.02 | 96.94 ± 0.20 | 96.28 ± 0.09 |
> > > | 30    | 97.53 ± 0.07  | 95.43 ± 0.38 | 94.90 ± 0.10 | 99.43 ± 0.03    | 96.46 ± 0.33    | 96.51 ± 0.06    | 98.92 ± 0.02 | 96.88 ± 0.10 | 96.31 ± 0.09 |
> > > | 50    | 97.57 ± 0.11  | 95.43 ± 0.26 | 94.95 ± 0.14 | 99.47 ± 0.04    | 96.43 ± 0.38    | 96.57 ± 0.16    | 98.92 ± 0.01 | 96.89 ± 0.11 | 96.32 ± 0.11 |
> > > | 70    | 97.59 ± 0.07  | 95.44 ± 0.34 | 94.98 ± 0.12 | 99.47 ± 0.05    | 96.41 ± 0.42    | 96.48 ± 0.08    | 98.92 ± 0.02 | 96.89 ± 0.13 | 96.31 ± 0.08 |
> > > | 100   | 97.62 ± 0.09  | 95.47 ± 0.35 | 95.01 ± 0.16 | 99.49 ± 0.05    | 96.38 ± 0.41    | 96.60 ± 0.13    | 98.92 ± 0.02 | 96.87 ± 0.11 | 96.32 ± 0.10 |

---

> > > > ### Comment · Reviewer_CKem · 2025-08-06
> > > > **Response to Authors' Comment**
> > > >
> > > > Thanks a lot for these additional results. I will update my "Quality" score accordingly.

---

### Official Review · Reviewer_rwYH · 2025-07-03

**Clarity:** 2
**Significance:** 2
**Originality:** 3
**Rating:** 3
**Confidence:** 3

**Summary:**

This paper introduces retain-free machine unlearning framework, RUAGO. It utilizes adversarial probability module to generate soft labels to mitigate over-unlearning and a OOD data generation model to distill the original model's knowledge to unlearned model. RUAGO generally outperforms existing retain-free unlearning methods and achieves comparable performance to retain-based methods, demonstrating its effectiveness.

**Questions:**

- Although it outperforms other methods in instance-wise unlearning, it seems to be not effective in class-wise unlearning for some cases. What makes this difference? Is there any analysis or ablation study for class-wise unlearning?

**Ethical Concerns:**

["NO or VERY MINOR ethics concerns only"]

**Final Justification:**

While the hyperparameter issues were clarified, the method’s limited applicability and reliance on a pretrained model remain.

Therefore, the rating remains unchanged at borderline reject.

**Limitations:**

- Although the authors theoretically have proven the generalization bound, it is limited to the curriculum learning, not to each method.

**Quality:**

2

**Strengths And Weaknesses:**

### Strength:
- This paper proposes retain-free unlearning framework which is effective for instance-wise unlearning.
- This paper provides theoretical analysis, as well as empirical results for the proposed methods.

### Weakness:
- Proposed methods seem to be applicable on the limited situations, for example, this paper only considers image classification. Moreover, one term in L_inv is only applicable to the batchnorm-based model.
- Proposed methods introduce additional hyperparameters, such as γ_cls, γ_adv, and γ_bn, but there is no analysis about the effect of each hyperparameter.
- Pretrained generative model is necessary. Unlearning performances are highly degrades without it.

---

> ### Author Rebuttal · Authors · 2025-07-31
>
> **We thank Reviewer rwYH for the thoughtful comments.**
>
> **[W1] Limited applicability.**
>
> We appreciate the reviewer’s comment. Our current scope is image classifiers under a retain-free constraint. We chose this setting because it lets us analyze the basic unlearning mechanics. However, we believe that the fundamental design principles of RUAGO, particularly curriculum-based training over OOD samples, adversarial soft-target regularization, and generator-teacher alignment, are not inherently tied to our current domain.
>
> For instance, in NLP settings, one could replace the generator with a small language model, define sample difficulty using token-level cross-entropy, and implement alignment in the hidden feature space instead of the pixel space. Similarly, adversarial soft-targets could be constructed via perturbations applied to token logits.
>
> We do not claim that our method can be directly applied to generative models. Adapting RUAGO to GANs, diffusion models, or LLMs would require careful redesign and task-specific metrics. In the appendix (Lines 1147–1149), we explicitly note that extending RUAGO beyond supervised image classification is still to be verified. In short, this work establishes the method in the simplest supervised setting. Extending to other domains is feasible but outside our present scope. We will add a brief roadmap in the Discussion to clarify.
>
> Regarding the final term in $\mathcal{L}\_{\text{inv}}$, this batch-norm statistics regularizer applies only to BN architectures. We use it to speed convergence by steering generator outputs toward natural feature statistics during inversion. In our ViT experiments (which lack batch normalization), we simply disabled this term by setting $\gamma\_{\text{bn}} = 0$, and RUAGO still outperformed other retain-free methods in both UA and TA (see Table 1). Its usage is similar to that in recent DFKD studies, which also employ batchnorm cues to improve synthetic sample quality. We will clarify in the final version that this regularization is optional and safely deactivated in architectures without batch normalization.
>
> **[W2] Experimental complexity.**
>
> We appreciate the reviewer’s comment regarding multiple objectives and the broad hyperparameter space. We adopted a broad search range to capture strong configurations across architectures and settings. To address the concern about multiple objectives and a broad hyperparameter space, we conducted a sensitivity analysis of the loss weights and examined how each term affects performance.
> In each experiment, we varied a single hyperparameter while fixing others to isolate its individual effect. On the VGGFace2 dataset with the ResNet-18 model, we varied $\gamma_{\text{adv}}, \gamma_{\text{bn}}, \gamma_{\text{cls}}$ in the range of [0, 5] as shown below:
>
> | $\gamma_{\text{adv}}$ | $\gamma_{\text{bn}}$ | $\gamma_{\text{cls}}$ | RA      | UA    | TA    |
> | --------- | -------- | --------- | ------- | ----- | ----- |
> | 0         | 1        | 1         | 99.2490 | 96.8387 | 96.2874 |
> | 1         | 1        | 1         | 99.2564 | 96.9228  | 96.2734 |
> | 2         | 1        | 1         | 99.2434 | 96.8556 | 96.2874 |
> | 3         | 1        | 1         | 99.2378 | 96.8387 | 96.2171  |
> | 4         | 1        | 1         | 99.2471 | 96.8556 | 96.2593 |
> | 5         | 1        | 1         | 99.2359 | 96.8219 | 96.2312  |
> | 0         | 2        | 1         | 99.2546 | 96.9228 | 96.2453 |
> | 0         | 3        | 1         | 99.2471 | 96.8556 | 96.2312 |
> | 0         | 4        | 1         | 99.2583 | 96.8387 | 96.2734 |
> | 0         | 5        | 1         | 99.2471 | 96.8556 | 96.2312 |
> | 0         | 1        | 2         | 99.2490 | 96.8219 | 96.2593 |
> | 0         | 1        | 3         | 99.2378 | 96.8219 | 96.2312 |
> | 0         | 1        | 4         | 99.2471 | 96.8892 | 96.2874 |
> | 0         | 1        | 5         | 99.2359 | 96.8892 | 96.2734 |
>
> The results show that RA, UA, and TA remain stable with variations limited to within ±1%. This indicates that the performance is not overly sensitive to individual loss and can maintain robust performance across a wide range of hyperparameter settings.
> Additionally, we conducted further experiments to evaluate the effect of varying $\gamma_1$ and $\gamma_2$:
>
> | $\gamma_1$ | $\gamma_2$ | RA      | UA      | TA      |
> | ------- | ------- | ------- | ------- | ------- |
> | 0       | 0.01    | 95.3425 | 95.2917 | 91.6467 |
> | 0.15    | 0.01    | 99.2359 | 96.8556 | 96.2874 |
> | 0.5     | 0.01    | 99.2789 | 96.4520  | 96.3859 |
> | 1       | 0.01    | 99.2658 | 96.3511 | 96.3437 |
> | 1.5     | 0.01    | 99.2228 | 96.2838 | 96.3156 |
> | 0.15    | 0       | 98.5895 | 95.998  | 95.4859 |
> | 0.15    | 0.01    | 99.2359 | 96.8556 | 96.2874 |
> | 0.15    | 0.5     | 98.5353 | 98.907  | 95.3031 |
> | 0.15    | 1       | 97.9169 | 98.2176 | 94.4874 |
> | 0.15    | 1.5     | 97.4909 | 97.7131 | 93.8968 |
>
>
> When fixing $\gamma_2=0.01$ and varying $\gamma_1$ from 0 to 1.5, we consistently observed strong and stable performance. Fixing $\gamma_1=0.15$ and increasing $\gamma_2$ showed that excessively large $\gamma_2$ can slightly degrade performance.
>
> Despite involving multiple objective terms, these results demonstrate that RUAGO can maintain consistent and stable performance across a wide range of parameter values. This clearly shows that our method is robust and feasible even without extensive tuning, alleviating the concerns of the reviewer. As a practical guideline, in our experiments, we first fixed $\gamma_{adv}=\gamma_{bn}=\gamma_{cls}=1$ and $\gamma_{2}=0.01$. We then tuned $\gamma_{1}$ on a grid from 0.1 to 1.5. This one-dimensional sweep was sufficient to reach near-optimal UA and RA on every model–dataset pair, showing that RUAGO requires minimal effort for setting hyperparameters. We will describe this practical guideline explicitly in the final version.
>
> **[W3] Requiring pretrained generative model.**
>
> We thank the reviewer for the comments. As noted, RUAGO relies on a pretrained generator to simulate retain-like data in the retain-free setting. This dependency reflects a fundamental challenge of unlearning without access to the original data; therefore, some mechanism is needed to approximate the distribution that should be retained. Among several options, we chose an OOD-trained generator as an efficient and straightforward surrogate. As shown in Table 7, using an untrained generator results in significant performance drops, demonstrating the importance of a pretrained generator.
> However, pretraining the generator is not necessarily costly. In our experiments, G was trained on publicly available OOD datasets (e.g., COCO, TinyImageNet) for only 3–4 hours using StyleGAN2-ADA. We see this as a practical cost-performance tradeoff for the retain-free constraint.
> One possible approach to mitigate this dependency is to utilize off-the-shelf generative models, assuming their training data does not cause other privacy issues. We will clarify this limitation and potential extensions in the Limitation and Discussion sections of the final version.
>
> **[Q1] Analysis in class-wise unlearning results.**
>
> As the reviewer points out, in the class-wise setting, we observe that our approach did not perform the best in ResNet-18, while we achieved good performance  in other settings. We attribute this to the fact that RUAGO’s main signal, the adversarial soft target derived from the forget set, is designed to reshape decision boundaries locally around individual samples. In class-wise unlearning, the objective is to erase the entire concept of a target class. This requires driving its logits (and thus predicted probabilities) toward zero everywhere in the input space, rather than only reshaping local decision boundaries.
> To keep the unlearning pipeline consistent across scenarios, we applied the same soft-label curriculum to both instance-wise and class-wise scenarios, and this led to residuals in a few model-setting combinations. Even so, on VGG-16 and ViT, the class-wise UA/TUA values were near zero, demonstrating near-complete deletion (unlearning performance), while RA/TRA (utility) remained stable.
> We also note that some retain-free baselines that performed well in class-wise deletion, such as SSD and SCAR, tend to struggle with instance-wise unlearning or with larger models and training budgets. In contrast, RUAGO offers balanced and robust performance across both scenarios.
> We will clarify this in the final version.
>
> **[L1] Limited theoretical analysis for all component.**
>
> Thank you for the comment. The theoretical bound we provide is indeed limited to the curriculum learning aspect. We chose this focus to analyze how the difficulty and hypothesis space complexity evolve during training, which is essential for stabilizing distillation from OOD-generated samples. This analysis helps explain why RUAGO achieves strong UA and TA without the retain data.
> We agree that a unified theoretical treatment encompassing all components of RUAGO has not yet been established. We consider this as an important direction for future work and will note this limitation, along with our plan for integrated analysis, in the final version.

---

> > ### Comment · Reviewer_rwYH · 2025-08-03
> >
> > Thank you for the thorough rebuttal. While several issues were clarified, the limited applicability and its reliance on a pretrained model remain problematic, so the rating remains unchanged.

---

### Official Review · Reviewer_Joyp · 2025-07-08

**Clarity:** 1
**Significance:** 2
**Originality:** 3
**Rating:** 4
**Confidence:** 4

**Summary:**

The paper proposes Retain-free Unlearning via Adversarial attack and Generative model using OOD training (RUAGO), a novel retain-free image classifier unlearning method that does not require retain-data to prevent over-unlearning and preserve post-unlearning utility. RUAGO achieves this by: (1) generating adversarial soft targets from the forget set to guide the model’s previously learned decision boundaries, and (2) training a generative model on out-of-distribution (OOD) data to produce synthetic data, which can be used in place of retain-data without privacy concerns. The paper supports the use of synthetic data with a VC-theoretic analysis that informs a dynamic sample difficulty scheduler, and also employs inversion-based fine-tuning to align the generative model’s output distribution with the original data. RUAGO is evaluated against existing unlearning methods on the CIFAR-10/100, TinyImageNet, and VGGFace2 datasets using the VGG16, ResNet18, and ViT architectures, demonstrating strong empirical performance.

**Questions:**

- [Q1] Could Figure 1 be revised to make it more understandable? Specifically, what does "G" mean in retain-free w/ "G" and G trained w/ OOD? Could Baseline 1 & 2 also be specified?

**Ethical Concerns:**

["NO or VERY MINOR ethics concerns only"]

**Final Justification:**

The paper offers a novel and interesting theoretical perspective on retain-free unlearning, and the results shared in the rebuttal demonstrate stable empirical performance. While I am leaning toward a positive score, I remain somewhat borderline due to concerns about its limited applicability.

**Limitations:**

The authors adequately addressed the limitations of their work.

**Quality:**

2

**Strengths And Weaknesses:**

## Strengths
- [S1] **Clear motivation and novelty.** Over-unlearning and the assumption of access to original retain-data are clear limitations in existing unlearning methods. A novel technique that resolves both would be a valuable addition to the unlearning toolkit. Incorporating VC theory to design a difficulty-based learning curriculum is also a novel idea in this context and could be of significant interest to the community.
- [S2] **Strong empirical performance.** Experimental results clearly show that RUAGO performs well in terms of both classification accuracy and privacy protection.

## Weaknesses
- [W1] **Narrow scope and applicability.** The applicability of RUAGO is currently limited to unlearning image classifiers. There does not appear to be a straightforward way to extend RUAGO to generative models (where privacy concerns are more pressing due to higher pretraining costs and the potential for data memorization) since many of RUAGO's components assume a classification setup.
- [W2] **Unclear theoretical implications.** Section 4.3 is quite unclear in its current form. For instance in Theorem 1, what does VC-dimension and $R(\mathcal{F})$ mean? How does Theorem 1 justify quantifying each sample’s difficulty using its loss, as in Equation 4? What assumptions underlie the generalization bound, and are they reasonable within the distillation setup? Several such pieces are missing, making it difficult to assess the theoretical contribution of the paper.
- [W3] **Experimental complexity.** The method involves multiple objective terms during training and requires extensive tuning of loss weights (e.g., the $\\gamma$ weights in Equation 5 and 6). The tested ranges of these hyperparameters also seem quite broad ($\\gamma_{cls}, \\gamma_{adv}, \\gamma_{bn} \\in [0, 5]$, $\\gamma_1 \\in [0.01, 1.50]$ and $\\gamma_2 = 0.01$). Could the authors provide guidance on how these values/ranges were chosen, how the results are expected to behave with different hyperparameter choices, and what the actual sensitivity of the method was?

---

> ### Author Rebuttal · Authors · 2025-07-31
>
> **Thank you for Reviewer Joyp's insightful comments on our work.**
>
> **[W1] Narrow scope and applicability.**
>
> We appreciate the reviewer’s comment. Our current scope is image classifiers under a retain-free constraint. We chose this setting because it lets us analyze the basic unlearning mechanics. However, we believe that the fundamental design principles of RUAGO, particularly curriculum-based training over OOD samples, adversarial soft-target regularization, and generator-teacher alignment, are not inherently tied to our current domain.
>
> For instance, in NLP settings, one could replace the generator with a small language model, define sample difficulty using token-level cross-entropy, and implement alignment in the hidden feature space instead of the pixel space. Similarly, adversarial soft-targets could be constructed via perturbations applied to token logits.
>
> We do not claim that our method can be directly applied to generative models. Adapting RUAGO to GANs, diffusion models, or LLMs would require careful redesign and task-specific metrics. In Appendix (Lines 1147–1149), we explicitly note that extending RUAGO beyond supervised image classification is still to be verified. In short, this work establishes the method in the simplest supervised setting. Extending to other domains is feasible but outside our present scope. We will add a brief roadmap in the Discussion to clarify.
>
>
> **[W2] Unclear theoretical implications.**
>
> Thank you for Reviewer Joyp’s thoughtful comments on the theoretical clarity. We clarify our responses to the three sub‑questions.
>
> **VC-dimension and risk notation.**
> The Vapnik–Chervonenkis (VC) dimension is the standard measure of a hypothesis space's capacity. In our setting, each curriculum stage $t$ explores a finite *effective* hypothesis space $\mathcal{H}\_t$ with VC‑dimension $d\_t=\mathrm{VCdim}(\mathcal{H}\_t)$. We denote the true risk by $R(\mathcal{F})=\mathbb{E}\_{(x,y)\sim\mathcal{D}}\bigl[\ell\bigl(\mathcal{F}(x),y\bigr)\bigr],$ and the empirical risk by $\hat{R}\_n(\mathcal{F})=\frac1n\sum_{i=1}^{n}\ell\bigl(\mathcal{F}(x\_i),y\_i\bigr).$
>
>
> **Curriculum assumption and Theorem 1.**
> Theorem 1 assumes the following: at early stages (easy samples) $\mathcal{H}\_t$ is small so $d\_t$ is small; as harder samples are added, $d\_t$ grows gradually. By progressively controlling this stage‑wise capacity, Theorem 1 upper‑bounds the risk of the unlearned model $\mathcal{F}\_\theta^{U}$ by a finite margin concerning the retrain (gold‑standard) model $\mathcal{F}\_{\theta^*}$. A closely related VC-based bound is used in recent data-free KD work, such as CuDFKD [47], which explicitly bounds each stage's risk via the VC dimension; our distillation pipeline is similar to these assumptions.
>
>
> **Loss‑based difficulty scheduling.**
> We implement the curriculum by defining sample difficulty as a monotonic decreasing function of the current loss, $w_k=\frac{1+\exp(-1/\lambda)}{1+\exp(\,\ell_k-1/\lambda\,)},$ and gradually increasing $\lambda$ so that training starts with easy (low‑loss) samples and progressively focuses on harder ones. OOD synthetic training data can introduce noise and distribution shift. Accordingly, we initialize with easy samples to keep the effective capacity and empirical risk small, which helps prevent divergence when harder samples are increasingly considered. We establish this connection immediately after Theorem 1 to demonstrate how the VC-bound justifies Eq. (4) and our easy-to-hard strategy, thereby explaining why the model converges toward the desired unlearning solution.
> We hope this will clarify the ambiguity in W2 and explain the theoretical contribution clearly.
>
> **[W3] Experimental complexity.**
>
> We appreciate the reviewer’s comment regarding multiple objectives and the broad hyperparameter space. We adopted a broad search range to capture strong configurations across architectures and settings. To address the concern about multiple objectives and a broad hyperparameter space, we conducted a sensitivity analysis of the loss weights and examined how each term affects performance.
> In each experiment, we varied a single hyperparameter while fixing others to isolate its individual effect. On the VGGFace2 dataset with the ResNet-18 model, we varied $\gamma_{\text{adv}}, \gamma_{\text{bn}}, \gamma_{\text{cls}}$ in the range of [0, 5] as shown below:
>
> | $\gamma_{\text{adv}}$ | $\gamma_{\text{bn}}$ | $\gamma_{\text{cls}}$ | RA      | UA    | TA    |
> | --------- | -------- | --------- | ------- | ----- | ----- |
> | 0         | 1        | 1         | 99.2490 | 96.8387 | 96.2874 |
> | 1         | 1        | 1         | 99.2564 | 96.9228  | 96.2734 |
> | 2         | 1        | 1         | 99.2434 | 96.8556 | 96.2874 |
> | 3         | 1        | 1         | 99.2378 | 96.8387 | 96.2171  |
> | 4         | 1        | 1         | 99.2471 | 96.8556 | 96.2593 |
> | 5         | 1        | 1         | 99.2359 | 96.8219 | 96.2312  |
> | 0         | 2        | 1         | 99.2546 | 96.9228 | 96.2453 |
> | 0         | 3        | 1         | 99.2471 | 96.8556 | 96.2312 |
> | 0         | 4        | 1         | 99.2583 | 96.8387 | 96.2734 |
> | 0         | 5        | 1         | 99.2471 | 96.8556 | 96.2312 |
> | 0         | 1        | 2         | 99.2490 | 96.8219 | 96.2593 |
> | 0         | 1        | 3         | 99.2378 | 96.8219 | 96.2312 |
> | 0         | 1        | 4         | 99.2471 | 96.8892 | 96.2874 |
> | 0         | 1        | 5         | 99.2359 | 96.8892 | 96.2734 |
>
> The results show that RA, UA, and TA remain stable with variations limited to within ±1%. This indicates that the performance is not overly sensitive to individual loss and can maintain robust performance across a wide range of hyperparameter settings.
> Additionally, we conducted further experiments to evaluate the effect of varying $\gamma_1$ and $\gamma_2$:
>
> | $\gamma_1$ | $\gamma_2$ | RA      | UA      | TA      |
> | ------- | ------- | ------- | ------- | ------- |
> | 0       | 0.01    | 95.3425 | 95.2917 | 91.6467 |
> | 0.15    | 0.01    | 99.2359 | 96.8556 | 96.2874 |
> | 0.5     | 0.01    | 99.2789 | 96.4520  | 96.3859 |
> | 1       | 0.01    | 99.2658 | 96.3511 | 96.3437 |
> | 1.5     | 0.01    | 99.2228 | 96.2838 | 96.3156 |
> | 0.15    | 0       | 98.5895 | 95.998  | 95.4859 |
> | 0.15    | 0.01    | 99.2359 | 96.8556 | 96.2874 |
> | 0.15    | 0.5     | 98.5353 | 98.907  | 95.3031 |
> | 0.15    | 1       | 97.9169 | 98.2176 | 94.4874 |
> | 0.15    | 1.5     | 97.4909 | 97.7131 | 93.8968 |
>
>
> When fixing $\gamma_2=0.01$ and varying $\gamma_1$ from 0 to 1.5, we consistently observed strong and stable performance. Fixing $\gamma_1=0.15$ and increasing $\gamma_2$ showed that excessively large $\gamma_2$ can slightly degrade performance.
>
> Despite involving multiple objective terms, these results demonstrate that RUAGO can maintain consistent and stable performance across a wide range of parameter values. This clearly shows that our method is robust and feasible even without extensive tuning, alleviating the concerns of the reviewer. As a practical guideline, in our experiments, we first fixed $\gamma_{adv}=\gamma_{bn}=\gamma_{cls}=1$ and $\gamma_{2}=0.01$. We then tuned $\gamma_{1}$ on a grid from 0.1 to 1.5. This one-dimensional sweep was sufficient to reach near-optimal UA and RA on every model–dataset pair, showing that RUAGO requires minimal effort for setting hyperparameters. We will describe this practical guideline explicitly in the final version.
>
>
> **[Q1] Ambiguity in Figure 1.**
>
> We thank the reviewer for pointing out the ambiguity in Figure 1. In the final version, we will revise both the legend and the caption to improve clarity. Specifically, the "G" in "retain-free w/ G" refers to a generator trained on OOD data. This setting denotes the scenario where unlearning is performed solely using synthetic samples from this generator, without access to any retain data. Additionally, we will clarify that "Baseline 1" and "Baseline 2" refer to prior unlearning methods: Baseline 1 corresponds to Bad Teacher [17], which uses the retain set, and Baseline 2 to Boundary Shrink [19], which does not require retain data. We will revise these explicitly.

---

### Decision · Program_Chairs · 2025-09-17

**Decision:**

Accept (poster)

**Comment:**

This paper proposes RUAGO, a retain-free unlearning framework that combines adversarial soft labels, OOD-pretrained generative models, progressive sampling, and inversion-based alignment to remove sensitive data while maintaining model utility. The approach is motivated by the limitations of existing methods that either assume access to retain data or risk over-unlearning.

Reviewers agreed that the motivation is clear and that the paper addresses an important problem, with strong empirical results showing RUAGO outperforming existing retain-free approaches and approaching retain-based baselines. One reviewer also highlighted the novelty of incorporating VC theory into the design of a difficulty-based curriculum. The paper is clearly written, and the rebuttal provided helpful clarifications, particularly on hyperparameters and theoretical details. Nonetheless, concerns remain regarding the limited applicability of the method.

Overall, the reviews reflect a mixed perspective: one borderline accept, one borderline reject, and one accept. Balancing the novelty, empirical strength, and open concerns, the AC finds the contribution valuable and recommends Accept.